# The conserved genetic program of male germ cells uncovers ancient regulators of human spermatogenesis

Rion Brattig-Correia[1,2†], Joana M Almeida[1,3†], Margot Julia Wyrwoll[4†], Irene Julca[5], Daniel Sobral[6,7], Chandra Shekhar Misra[1,8], Sara Di Persio[9], Leonardo Gastón Guilgur[1], Hans-Christian Schuppe[10], Neide Silva[1], Pedro Prudêncio[11], Ana Nóvoa[1], Ana S Leocádio[1], Joana Bom[1], Sandra Laurentino[9], Moises Mallo[1], Sabine Kliesch[9], Marek Mutwil[5], Luis M Rocha[1,2‡], Frank Tüttelmann[4‡], Jörg D Becker[1,8*‡], Paulo Navarro-Costa[1,3*‡§]

[1]Instituto Gulbenkian de Ciência, Oeiras, Portugal; [2]Department of Systems Science and Industrial Engineering, Binghamton University, New York, United States; [3]EvoReproMed Lab, Environmental Health Institute (ISAMB), Associate Laboratory TERRA, Faculty of Medicine, University of Lisbon, Lisbon, Portugal; [4]Centre of Medical Genetics, Institute of Reproductive Genetics, University and University Hospital of Münster, Münster, Germany; [5]School of Biological Sciences, Nanyang Technological University, Singapore, Singapore; [6]Associate Laboratory i4HB - Institute for Health and Bioeconomy, NOVA School of Science and Technology, NOVA University Lisbon, Lisbon, Portugal; [7]UCIBIO - Applied Molecular Biosciences Unit, Department of Life Sciences, NOVA School of Science and Technology, NOVA University Lisbon, Caparica, Portugal; [8]Instituto de Tecnologia Química e Biológica António Xavier, Universidade Nova de Lisboa, Oeiras, Portugal; [9]Centre of Reproductive Medicine and Andrology, University Hospital Münster, Münster, Germany; [10]Clinic of Urology, Pediatric Urology and Andrology, Justus-Liebig-University, Giessen, Germany; [11]Instituto de Medicina Molecular João Lobo Antunes, Faculdade de Medicina, Universidade de Lisboa, Lisboa, Portugal

*For correspondence:
jbecker@itqb.unl.pt (JDB);
navarro-costa@medicina.ulisboa.pt (PN-C)

†These authors contributed equally to this work
‡These authors also contributed equally to this work

§Further information and requests for resources and reagents should be directed to and will be fulfilled by the lead contact, Paulo Navarro-Costa (navarro-costa@medicina.ulisboa.pt)

**Abstract** Male germ cells share a common origin across animal species, therefore they likely retain a conserved genetic program that defines their cellular identity. However, the unique evolutionary dynamics of male germ cells coupled with their widespread leaky transcription pose significant obstacles to the identification of the core spermatogenic program. Through network analysis of the spermatocyte transcriptome of vertebrate and invertebrate species, we describe the conserved evolutionary origin of metazoan male germ cells at the molecular level. We estimate the average functional requirement of a metazoan male germ cell to correspond to the expression of approximately 10,000 protein-coding genes, a third of which defines a genetic scaffold of deeply conserved genes that has been retained throughout evolution. Such scaffold contains a set of 79 functional associations between 104 gene expression regulators that represent a core component of the conserved genetic program of metazoan spermatogenesis. By genetically interfering with the acquisition and maintenance of male germ cell identity, we uncover 161 previously unknown spermatogenesis genes and three new potential genetic causes of human infertility. These findings emphasize the importance of evolutionary history on human reproductive disease and establish a cross-species analytical pipeline that can be repurposed to other cell types and pathologies.

## eLife assessment

This **fundamental** study reports the deep evolutionary conservation of a core genetic program regulating spermatogenesis in flies, mice, and humans. **Convincing** data were presented and supported the main conclusion. This work will be of interest to evolutionary and reproductive biologists.

## Introduction

Understanding what defines the uniqueness of a given cell type out of the 843 predicted cellular fates in the human body is a complex and fascinating problem (*Han et al., 2020*). Through Conrad Waddington's foundational work, we have come to appreciate that developmental trajectories ultimately dictate cell type identity via the establishment of specific transcriptional programs (*Hermann et al., 2018*). The fact that transcriptomes tend to cluster by tissue type rather than by species (*Merkin et al., 2012*) clearly indicates that gene expression identity can be maintained across many million years of evolutionary divergence. This echoes the modular nature of eukaryotic biological processes, whose intervening macromolecular complexes are typically built by the addition of younger components to a core block of ancient subunits (*Wan et al., 2015*).

The emergence of germ cells is considered one of the first cell type specializations in metazoan history (*Arendt, 2008*). Since the capability to undergo both sexual reproduction and gametogenesis were already present in the unicellular ancestor of all Metazoa (*Sebé-Pedrós et al., 2017*), the split between germ line and soma presumably provided early multicellular organisms increased robustness against mutations while minimizing genetic conflict between different cell lineages (*Extavour, 2007*). Yet, reproductive protein genes are some of the most rapidly evolving across the tree of life (*Swanson and Vacquier, 2002*), leading to substantial variation in male reproductive tissues even between closely related species (*Ramm et al., 2014*; *Fitzpatrick et al., 2022*). At the molecular level, this variation can be mainly traced back to rapid evolutionary changes in gene expression in the testis (*Cardoso-Moreira et al., 2019*) – likely facilitated by widespread transcriptional leakage in male germ cells (*Soumillon et al., 2013*) – and to the preferential emergence of new genes in this organ (*Kaessmann, 2010*), where they often acquire functions in spermatogenesis (*Kondo et al., 2017*). Although the rapid divergence of genes directly involved in reproduction has traditionally been interpreted from an adaptationist stance, relaxed selection and drift have been recently proposed to account for this pattern (*Dapper and Wade, 2020*). Central to this debate is the often-overlooked contribution of old genes for germ cell development and function.

We have recently observed, in a wide range of plant species, a substantial contribution of old genes to the pollen transcriptome, suggestive of an ancient transcriptional program common to plant male gametes (*Julca et al., 2021*). Such observation finds parallel in the concept of a metazoan germ line multipotency program (*Fierro-Constaín et al., 2017*), and is supported by transcriptional similarities between equivalent spermatogenic cell types across mammalian species (*Lau et al., 2020*; *Shami et al., 2020*; *Murat et al., 2023*). This led us to explore the possibility that, despite the rapid evolution of reproductive protein genes, the functional basis of metazoan male germ cell identity relies on an old, evolutionarily conserved genetic program that can provide relevant insight into spermatogenesis and human infertility.

To test this hypothesis, we devised an interdisciplinary research platform based on four combined approaches (*Figure 1a*). First, we determined, through comparison of genome sequences, the age of the gene expression program of male germ cells from three evolutionarily distant metazoan species: humans (*Homo sapiens*), mice (*Mus musculus*), and fruit flies (*Drosophila melanogaster*). Then, we used network science to infer the significance of deeply conserved genes within the context of the complex male germ line transcriptome. Subsequently, through developmental biology (in vivo RNAi in fruit fly testes), we defined the role of a key subset of the conserved germ cell transcriptome in male reproductive fitness. Finally, we merged this information with clinical genetics to identify new potential causes of human infertility. Overall, we show that deeply conserved genes play a prominent role in male germ cell regulation and that the disruption of this ancient genetic program leads to human reproductive disease.

**eLife digest** Sperm are one of the most remarkable cells in nature, safely housing genetic information while also often moving through foreign environments in search of an egg to fertilize. Central for sexual reproduction, sperm cells of all shapes and sizes are found in animals, plants and even some species of fungi. You may be familiar with the streamlined structure of human sperm, for example, with its round head and flexible tail; but the sperm cells of fruit flies are about 300 times longer, and those found in mice have a hook-shaped head. Relatedly, the genes involved in the creation of reproductive cells often show rapid evolution, with their sequences quickly diverging between species. Due to the complexity of the network of genetic interactions taking place during sperm development, it has so far been difficult to fully isolate the 'core program' that governs sperm assembly and allows these cells to acquire their distinct identity. Whether this program could be conserved and shared across the tree of life, in particular, remains unclear.

In response, Brattig-Correia, Almeida, Wyrwoll et al. first conducted analyses that allowed them to pinpoint the genes that were 'switched on' during the formation of human, mouse and fruit fly sperm. Assessing the 'age' of these genes showed that a large proportion had emerged early during evolution. Shared across the three species, these deeply conserved genes were shown to play a fundamental role in sperm cells acquiring and maintaining their identity.

Further genetic experiments were conducted in fruit flies to refine these findings, highlighting a set of 161 previously unknown genes essential for sperm formation. By combining these results with genetic data from men unable to have children, Brattig-Correia, Almeida, Wyrwoll et al. were able to identify three new genes that could play a role in human infertility.

This work emphasizes how our understanding of human reproductive development can benefit from examining this process in other species, and its evolutionary history. In particular, the knowledge gained from these comparative approaches could ultimately help develop better genetic tests and treatments for human infertility.

## Results

### A more conservative estimate of the complexity of the male germ cell transcriptome

Male germ cell development is divided into three conserved stages (*Navarro-Costa et al., 2020*). The first is the pre-meiotic stage and corresponds to the mitotic expansion of committed precursors (spermatogonia). Meiosis defines the second stage, with the newly differentiated spermatocytes undergoing reductive division. In the third stage, the post-meiotic cells (spermatids) embark on a cytodifferentiation program that culminates in the formation of mature male gametes.

To understand to what extent male germ cell transcription quantitatively differs from that of somatic lineages, we collected previously published high-quality RNA-Seq datasets from pre-meiotic, meiotic, and post-meiotic germ cell populations, and compared them with representative somatic cell types of the primary embryonic layers: neurons (ectoderm), muscle (mesoderm), and enterocytes (endoderm; *Figure 1b* and *Supplementary file 1*). We centered our analysis on three evolutionarily distant gonochoric species with excellent genome annotations – humans, mice, and fruit flies (the last two also being well-established animal models for the functional study of human spermatogenesis) – and observed that, as previously reported (*Soumillon et al., 2013*; *Xia et al., 2020*), male germ cells have a generally more diversified transcriptome than their somatic counterparts (as measured by the percentage of the entire genome that each cell type expresses). Yet, this increased complexity was mainly restricted to extremely permissive minimum average expression cut-offs ranging from >0.01 to >0.5 transcripts per million (TPM), suggestive of a significant degree of transcriptional noise or other previously hypothesized translation-independent roles for lowly expressed transcripts (*Liu and Zhang, 2020*). Indeed, the use of a more suitable threshold for protein-coding transcripts (TPM >1; *Thul et al., 2017*) revealed that, on average, metazoan male germ cells tend to express approximately 10,000 genes (9,468±498–12,195±1144, depending on the species), corresponding to half of the entire protein-coding genome (48.0–67.9%; *Figure 1b* and *Supplementary file 1*). In both humans and mice, this range is comparable to that observed in somatic cell types such as neurons and

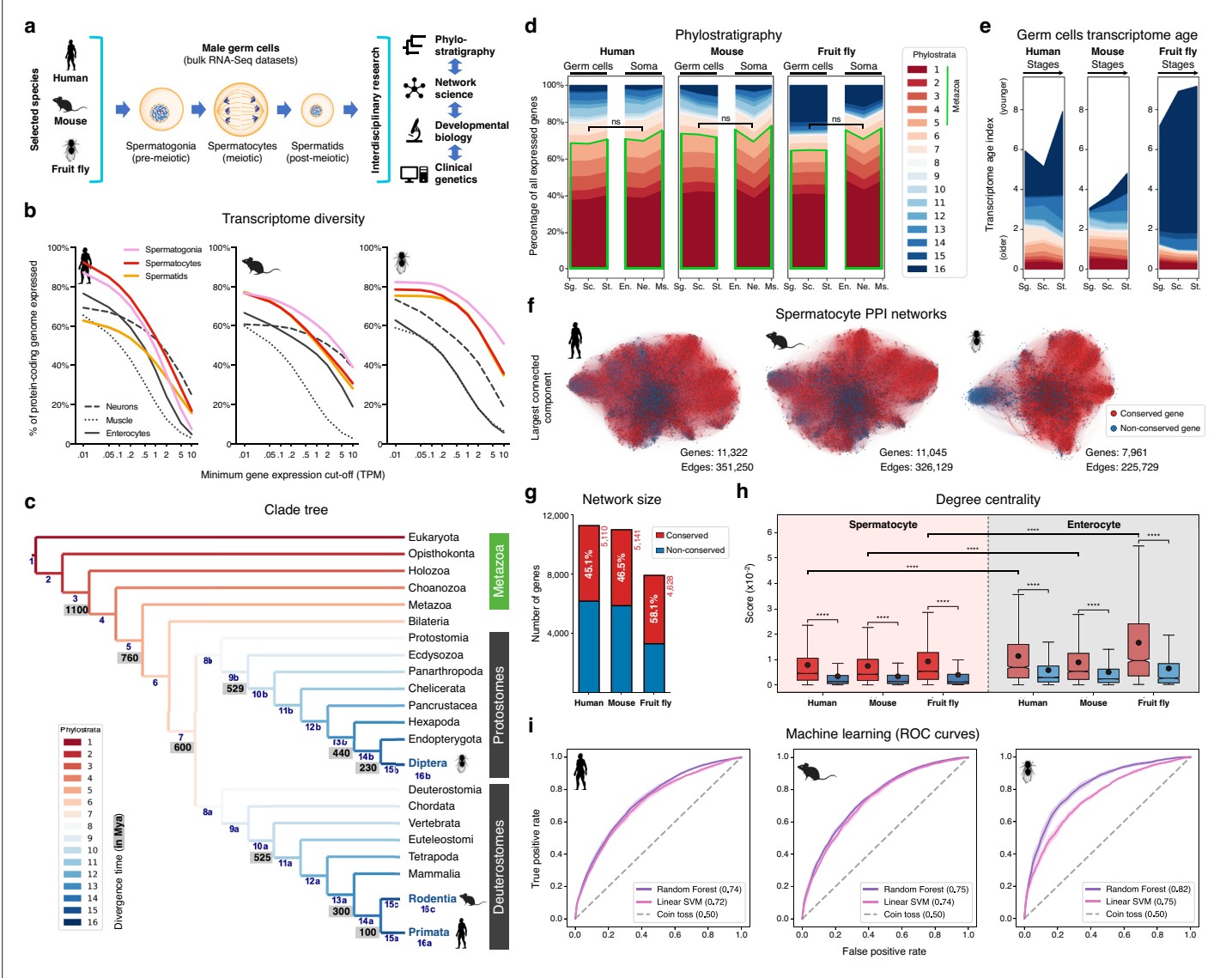

**Figure 1.** The male germ cell transcriptome has an old evolutionary origin. (**a**) Overview of the experimental strategy. (**b**) The diversity of the male germ cell transcriptome substantially depends on lowly-expressed genes. Three representative somatic cell types are included for comparison. TPM: transcripts per million; see *Supplementary file 1* for information on RNA-Seq datasets. (**c**) Clade tree for mapping the time of origin of genes in the three selected species: human (Primata), mouse (Rodentia), and fruit fly (Diptera). Genes assigned to phylostrata 1–5 are common to all metazoan species. Mya: million years ago; see *Figure 1—figure supplement 1* for the list of representative species of each clade and the number of genes in each phylostratum. (**d**) The majority of genes expressed by male germ cells are common to all Metazoa (phylostrata 1–5, green outline). This fraction is similar to that found in representative somatic cell types of each selected species. Minimum average expression cut-off: TPM >1. ns- no significant difference (p>0.3472; Mann–Whitney U test). Sg.: Spermatogonia, Sc.: Spermatocytes, St.: Spermatids, En.: Enterocytes, Ne.: Neurons and Ms.: Muscle. (**e**) Post-meiotic male germ cells have younger transcriptomes than meiotic and pre-meiotic cells. Transcriptome age indices (TAIs) are split between the different phylostrata. (**f**) The spermatocyte transcriptome forms a large, structured network based on protein-protein interaction (PPI) data. Graphs represent the largest connected component of all spermatocyte-expressed genes (minimum average expression cut-off: TPM >1) according to STRING functional association scores. Gene conservation (across all Metazoa) was defined based on eggNOG orthogroups. Networks were filtered to only include edges with combined scores ≥0.5 (see *Figure 1—figure supplement 3*). (**g**) Spermatocyte PPI networks contain a substantial number of conserved genes. (**h**) Conserved genes (red) are more connected than non-conserved genes (blue) in both germ cell (spermatocyte) and somatic cell (enterocyte) PPI networks ****p<0.0001 (Kolmogorov-Smirnov test). (**i**) Machine-learning algorithms reliably predict the evolutionary conservation of spermatocyte-expressed genes based solely on PPI network features. Values correspond to AUC (area under the curve) scores. 'Coin toss' corresponds to a random classification. Fourfold cross-validation results are shown. ROC: receiver operating characteristic curves; SVM: support-vector machine; see *Figure 1—figure supplement 5* for precision and recall curves.

The online version of this article includes the following figure supplement(s) for figure 1:

*Figure 1 continued on next page*

*Figure 1 continued*

**Figure supplement 1.** List of representative species of each clade for the phylostratigraphic analysis.

**Figure supplement 2.** Ubiquitously expressed genes in male germ cells.

**Figure supplement 3.** Filtering the protein-protein interaction (PPI) networks.

**Figure supplement 4.** Conserved genes have more connected interactors than non-conserved genes in protein-protein interaction networks.

**Figure supplement 5.** Precision and recall curves confirm the reliability of machine-learning algorithms in predicting evolutionary conservation of spermatocyte genes.

**Figure supplement 6.** Rewiring the spermatocyte protein-protein interaction networks.

enterocytes (53.4–53.7% and 37.3–45.6% of the genome, respectively). Only in fruit flies, a species with a smaller genome (13,947 protein-coding genes *vs.* 22,802 and 22,287 in humans and mice, respectively), the diversity of the male germ cell transcriptome diverged substantially from that found in representative somatic tissues (with an average of 67.9 ± 3.6% of the genome expressed in germ cells *vs.* 33.3 ± 12.5% in the soma). It should be noted that this percentage may be higher in fruit flies due to the presence of transcripts of somatic origin in the germ cell dataset (see *Supplementary file 1*). Collectively, these observations suggest that the average functional requirement of a metazoan male germ cell corresponds to the expression of approximately 10,000 protein-coding genes. Such a number brings a significant but largely overlooked functional constraint to the general notion that almost all protein-coding genes are transcribed in the male germ line.

## The male germ cell transcriptome has an old evolutionary origin

We next assessed the contribution of deeply conserved genes to the male germ cell transcriptome. For this we used phylostratigraphy, a technique that determines the evolutionary age of groups of homologous genes (defined as orthogroups in our analysis, including both orthologs and paralogs) by mapping their last common ancestor in a species tree (*Domazet-Loso et al., 2007*). We assembled our tree based on the proteomes of 25 phylogenetically representative eukaryotic species, and assigned each orthogroup to the oldest phylogenetic node (phylostratum) it could be traced back to (*Figure 1c* and *Figure 1—figure supplement 1*). This way, phylostrata ranked 1–5 were the oldest and contained orthogroups common to all Metazoa, while phylostratum 16 contained the youngest, species-specific orthogroups. A total of 113,757 orthogroups were identified in the 25 representative eukaryotes, 85.5% of which (97,270) were species-specific when adding all phylostrata. By using TPM >1 as minimum expression cut-off (to minimize the effects of widespread, low-level transcription in the male germ line), we observed that 48.0–67.9% of the entire protein-coding genome was expressed in male germ cells, depending on the species. Despite the rapid divergence typically associated with reproduction-related genes, we found that the majority (65.2 to 70.3%) of genes expressed in male germ cells mapped to the oldest-ranking phylostrata containing orthogroups common to all Metazoa, henceforth referred to as deeply conserved genes (*Figure 1d*). Indeed, in all three tested species, this percentage was not significantly different from that recorded in the transcriptome of the somatic cell types (also at a threshold of TPM >1). The fraction of deeply conserved genes that were ubiquitously expressed (proxy to their involvement in cellular housekeeping processes) varied from 49.7 to 63.4% (*Figure 1—figure supplement 2a, b*). This strongly suggests that almost half of all deeply conserved genes expressed by male germ cells are involved in more specific roles than just the maintenance of basal cellular functions.

By summing the products of the age of all expressed genes and their expression levels at a given developmental stage - a metric known as the transcriptome age index - TAI (*Domazet-Lošo and Tautz, 2010*), we determined the transcriptome age of the different male germ cell stages (*Figure 1e*). As recently reported in mammalian spermatogenesis (*Murat et al., 2023*), meiotic and pre-meiotic cells across all three tested species had lower TAIs than post-meiotic cells, indicative of older transcriptomes. This trend was less obvious in the fruit fly as in this species post-meiotic transcription is largely residual (*Lim et al., 2012*). Collectively, we observed that, both in vertebrates and invertebrates, the male germ cell transcriptome has an old evolutionary origin that is tempered by the increased expression of younger genes at later developmental stages. Our data, indicative of a core spermatogenic program spanning more than 600 million years of evolution, significantly expand the breadth of previous observations in the mammalian lineage (*Lau et al., 2020*; *Shami et al., 2020*; *Murat et al.,*

*2023*), while emphasizing the potential relevance of using conserved expression between distant animal species to gain insight into the molecular basis of human infertility.

## Deeply conserved genes are central components of the male germ cell transcriptome

We next addressed the possible significance of the abundant expression of deeply conserved genes in the male germ line. For this, we focused on meiosis, a fundamental process conserved through eukaryotes. Besides the existence of an ancient meiotic toolkit required for recombination and reduction division across sexually reproducing species (*Schurko and Logsdon, 2008*), the mitosis-to-meiosis transition in the male germ line is also home to a transcriptional burst associated with the acquisition of male germ cell identity (*Maezawa et al., 2020*). The transcripts produced in this burst are mainly required for post-meiotic development, with their translation being shifted towards the later spermatogenic stages (*Wang et al., 2020*). Accordingly, when analyzing the expression pattern of all human and mouse genes annotated to the 'spermatid differentiation' Gene Ontology (GO) term (GO:0048515), we observed that only a small fraction of these was expressed primarily/exclusively after meiosis (5.7% [13 out of 228] and 8.7% [25 out of 289] in humans and mice, respectively). Similar to previous reports in fruit flies (*Lim et al., 2012*; *Maezawa et al., 2020*; *Jan et al., 2017*), most of the mammalian spermatid differentiation genes were found to be expressed already at the meiotic stage, with 82.9% (189 out of 228, in humans) and 77.9% (225 out of 289, in mice) being expressed in spermatocytes or in spermatocytes and spermatogonia. These results show that the meiotic transcriptome provides a particularly suitable entry-point into the genetic basis of the core spermatogenic program. Hence, we assembled protein-protein interaction (PPI) networks based on the transcriptome data of human, mouse, and fruit fly spermatocytes, where nodes represent all expressed genes and edge weights indicate the probability of the connected genes contributing to a specific function according to the STRING database (*Szklarczyk et al., 2019*; *Figure 1f*). The structure of these networks reflects the multiple genetic interdependencies responsible for cellular function, as illustrated by the characteristic clustering of functionally related genes into topologically defined modules (*Choobdar et al., 2019*). Network edges were filtered to remove low confidence interactions (those with a combined confidence score <0.5, *Figure 1—figure supplement 3a–c*), and, as before, only genes expressed at TPM >1 were included to minimize the confounding effects of widespread low-level transcription. The resulting networks contained between 7961 and 11,322 genes, depending on the species, and an average of 301,036±66,416 edges.

Consistent with phylostratigraphy, the spermatocyte PPI networks included a substantial fraction of genes conserved across all Metazoa (45.1 to 58.1% of all genes; *Figure 1g*). As a starting point for our analyses, we set out to determine to what extent the previously reported increased connectivity of conserved genes in PPI networks (*Wuchty et al., 2003*; *Brown and Jurisica, 2007*) is maintained in a cell type characterized by rapid rates of evolution. For comparison purposes, we assembled, as before, PPI networks based on the transcriptome data of human, mouse, and fruit fly enterocytes (a somatic cell type with a comparably young transcriptome; *Figure 1—figure supplement 2b*). These somatic cell networks, albeit smaller due to lower transcriptome diversity in enterocytes (*Figure 1b*), were comparable in size to those of the spermatocytes, containing between 3,310 and 9,837 genes, depending on the species, and an average of 203,296±117,545 edges (*Figure 1—figure supplement 3a–d*). Similarly to what we observed in enterocytes, deeply conserved genes in the spermatocyte PPI networks were significantly more connected than their non-conserved counterparts (higher degree centrality), and their interactors were themselves more connected than those of non-conserved genes (higher page rank, Kolmogorov-Smirnov test for the two analyses; *Figure 1h* and *Figure 1—figure supplement 4*). Although the connectivity properties of the spermatocyte conserved genes were lower than those expressed in enterocytes, they were still salient enough for machine learning classifiers to predict reliably if a gene was conserved just based on spermatocyte network features. More specifically, using a Random Forest classifier, the AUC (area under the receiver operating characteristic curve) score was 0.74, 0.75, and 0.82 in the human, mouse, and fruit fly spermatocyte datasets, respectively (*Figure 1i*). Classification performance was equally high when using a different supervised learning algorithm (linear Support Vector Machine) and an alternative performance metric (precision and recall; *Figure 1—figure supplement 5*).

To address a possible ascertainment bias associated with more available information on conserved genes in the STRING database, we tested to what extent degree centrality and page rank were affected by network rewiring. In this approach, a variable percentage (20 to 100%) of all edges is randomly shuffled across the network, thus diluting any latent biases in the datasets. We observed that both network properties remained higher in conserved genes even when 80% of all edges were rewired in the spermatocyte PPI networks (*Figure 1—figure supplement 6*). Such result suggests that the increased connectivity of conserved genes is mostly driven by their intrinsic properties, rather than by differences stemming from the amount of source data available. Based on these analyses, we conclude that despite the testis being a rapidly evolving organ and a preferential birthplace for new genes, deeply conserved genes remain central components in the male germ cell transcriptome of evolutionarily distant species. Accordingly, we decided to explore this increased connectivity as a means of gaining insight into the conserved molecular basis of metazoan spermatogenesis.

## A core component of the ancient genetic program of spermatogenesis

We have previously shown that it is possible to simplify the complexity of biological networks by removing redundancy and only retaining the most relevant interactions that sustain the dynamics of the system (*Gates et al., 2021*). To do so, one can extract from a network its distance backbone, i.e., the small subset of metric edges that is sufficient to compute all shortest paths in the original network (*Simas et al., 2021*). This technique retains the minimum number of required edges to preserve all network nodes, as well as all of the natural hierarchy of connections and community structures (*Brattig Correia et al., 2023*). In the context of biological networks, such distance backbone uncovers the subgraph of gene interactions more likely to convey the key functional characteristics of the gene expression program (*Ren et al., 2018*).

By extending orthology to the backbone, we have developed a new type of multilayer network: the orthoBackbone. It expands on the concept of interologs (*Walhout et al., 2000*; *Sun and Kim, 2011*), as it consists of only the edges that connect the same pair of orthologs in the network backbone of different species (*Figure 2a*). Since this technique excludes all backbone edges (and corresponding nodes) that do not involve the same orthologs across multiple species, it ultimately identifies a subset of conserved genes whose functional interactions at the protein level are part of the distance backbone across evolution. In this regard, the orthoBackbone offers access to what can be considered core features of the genetic program of a given cellular state. Importantly, the use of fruit flies (an animal without male crossing over and recombination) in this inter-species approach significantly reduces the contribution of the ancient meiotic toolkit to the orthoBackbone, thus emphasizing functional interactions associated with spermatogenesis but not with meiosis per se.

We observed that the orthoBackbone drastically reduced the size of the spermatocyte PPI networks, retaining only 1.7 to 2.7% of all edges, depending on the species (*Figure 2b*). This almost residual fraction of edges nevertheless contained 70% of all deeply conserved genes in the network (*Figure 2c* and *Supplementary file 2*), further illustrating that strong functional relationships between deeply conserved genes are also maintained in the male germ line. GO term enrichment analysis for biological processes revealed that orthoBackbone genes were preferentially involved in gene expression and protein regulation, in contrast with the other (not retained) conserved genes that were mainly associated with cell signaling pathways (*Figure 2d* and *Figure 2—figure supplement 1*). To better understand to what extent the orthoBackbone can provide insight into cell-type specific processes, we determined the degree of similarity between the spermatocyte and enterocyte orthoBackbones. We noted a modest overlap between both subnetworks, with just 16.2% to 18.0% of common edges depending on the species (against background expectations of 33.6 to 61.5%, based on Jaccard similarity index scores between the PPI networks of both cell types; *Figure 1—figure supplement 3d*).

Using GO annotations, we further selected spermatocyte orthoBackbone interactions involving gene expression regulators previously linked to spermatogenesis. By doing so, we obtained a subset of 79 functional interactions between 104 human genes (out of the total of 3,596 in the orthoBackbone; *Figure 2e*). These were ascribed to a wide gamut of regulatory processes and involved noteworthy spermatogenesis genes such as *RFX2* (transcription), *CDYL* (chromatin remodeling) and *BOLL* (translation), among others (*Wu et al., 2016*; *Liu et al., 2017*; *Shah et al., 2010*). Of the 104 genes, 34 were testis-specific/enriched while 60 had low tissue specificity, consistent with previous observations that cell identity relies on the integration of cell type-specific genes with more broadly expressed

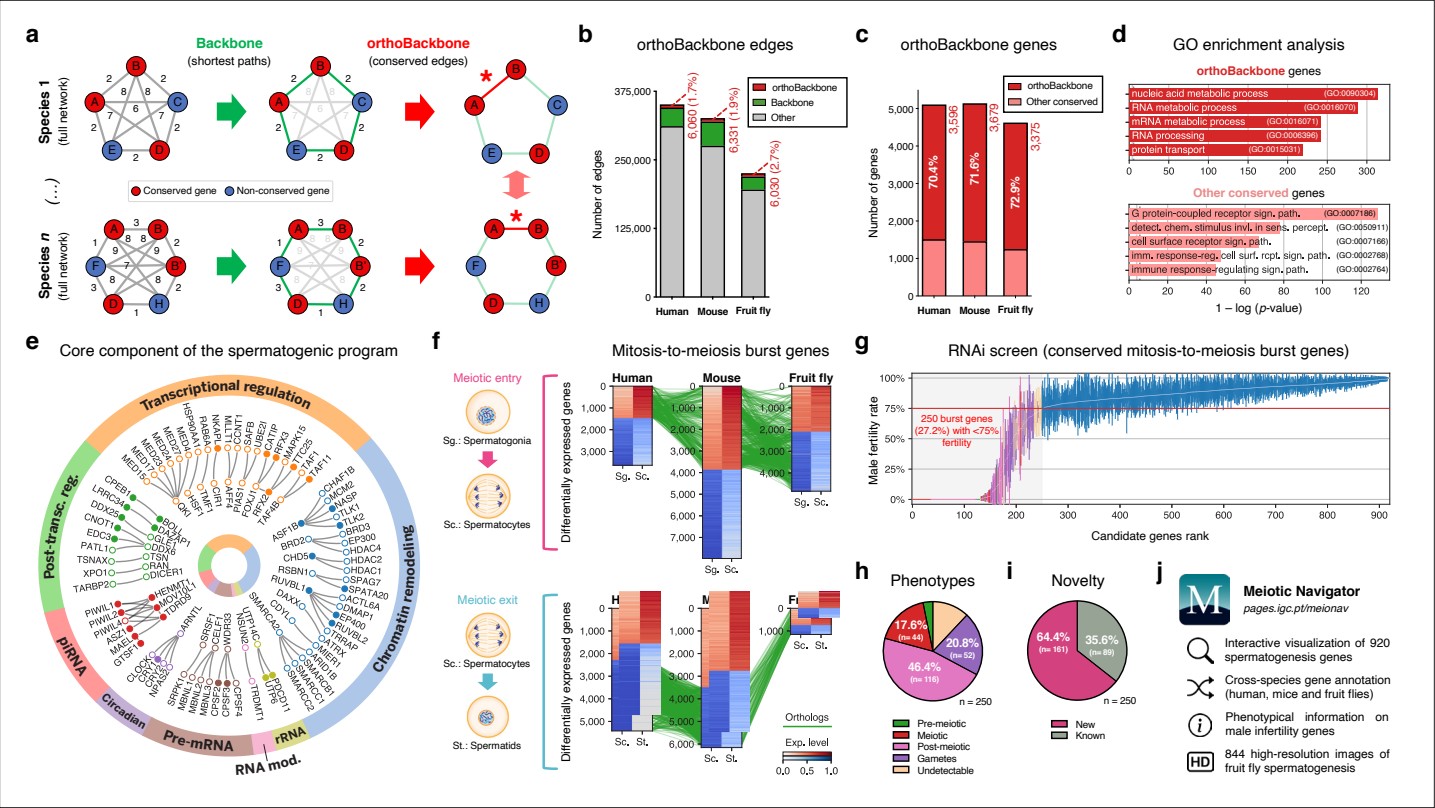

**Figure 2.** Functional analysis of the conserved genetic program of male germ cells. (**a**) The orthoBackbone methodology. First, the most relevant associations are determined by defining the individual metric backbones (based on shortest paths) of protein-protein interaction (PPI) networks from different species. Of the backbone edges (in green), those connecting the same orthologous genes across the different species are selected as part of the evolutionarily conserved orthoBackbone (in red, with asterisks). In the case of a one-to-many conserved edge relationship, inclusion depends on at least one of the multiple edges being part of the backbone. Letters depict different genes, B' and B" are paralogs, and numbers indicate distances between genes. (**b**) The orthoBackbone represents less than 3% of all functional interactions (edges) in the spermatocyte PPI networks. (**c**) The orthoBackbone connects >70% of all conserved genes expressed in spermatocytes. Gene conservation (across Metazoa) was defined based on eggNOG orthogroups. (**d**) orthoBackbone genes are preferentially involved in gene expression regulation compared with other equally conserved genes. Charts represent the top 5 terms of an unfiltered gene ontology (GO) enrichment analysis for biological processes of the human male germ cell orthoBackbone. False discovery rate ≤0.05; see **Figure 2—figure supplement 1** for the expanded GO analyses. (**e**) The male germ cell orthoBackbone reveals a core set of 79 functional interactions between 104 gene expression regulators of spermatogenesis. Solid dots indicate genes with testis specific/enriched expression. Post-transc. reg.: Post-transcriptional regulation; RNA mod.: RNA modification. (**f**) Conserved mitosis-to-meiosis transcriptional burst genes were defined based on their upregulation at mammalian meiotic entry and/or downregulation at meiotic exit. In both cases, genes also had to be expressed in insect spermatogenesis. Green lines link orthologs (920 in fruit flies, 797 in humans and 850 in mice) based on eggNOG orthogroups. Expression level in normalized absolute log(FPKM +1). (**g**) An in vivo RNAi screen in fruit fly testes uncovers the functional requirement of 250 conserved transcriptional burst genes (27.2%) for male reproductive fitness. Silencing of the 920 genes was induced at the mitosis-to-meiosis transition using the *bam*-GAL4 driver. Color-code for the recorded testicular phenotype as in "h". Results reflect a total of four independent experiments. Threshold for impaired reproductive fitness (red horizontal line) corresponds to a 75% fertility rate (>2 standard deviations of the mean observed in negative controls). (**h**) Conserved transcriptional burst genes are required for diverse spermatogenic processes. Testicular phenotypes of the 250 hits were defined by phase-contrast microscopy and assigned to five classes based on the earliest manifestation of the phenotype. (**i**) Transcriptional burst genes reveal 161 new, evolutionarily conserved regulators of spermatogenesis (64.4% of all hits, homologous to 179 and 187 in humans and mice, respectively). Phenotype novelty was defined by lack of previously published evidence of a role in male fertility/spermatogenesis in humans, mice or fruit flies. (**j**) All data acquired in this screen are freely available in the form of an open-access gene browser (Meiotic Navigator).

The online version of this article includes the following figure supplement(s) for figure 2:

**Figure supplement 1.** Top 10 terms of an unfiltered gene ontology (GO) enrichment analysis of the human male germ cell orthoBackbone.

regulators (*Kotliar et al., 2019*). Based on its conservation in the orthoBackbone of evolutionarily distant species and on the significant spermatogenic role of the intervening genes, we posit that this subnetwork of 79 functional interactions involving 104 genes likely represents a core component of the conserved genetic program of spermatogenesis. This number fits well with previous estimates,

based on expression trajectories of 1:1 orthologous genes, of an ancestral program of amniote spermatogenesis constituted by 389 genes (*Murat et al., 2023*).

## Male germ cell identity has a broad functional basis

We next set out to determine the functional consequences of disrupting the spermatocyte orthoBackbone. Since the latter represents a still sizeable interaction subnetwork consisting of more than 3,000 genes, we decided to select candidate genes likely to affect the acquisition and maintenance of male germ cell identity. For this, we again took into account how the mitosis-to-meiosis transcriptional burst activates the expression of spermatogenesis genes – a process that has been likened to that of cellular reprogramming (*Guo et al., 2018*; *Alavattam et al., 2019*). We reasoned that genes that are upregulated at meiotic entry and/or downregulated at meiotic exit (thus part of the transcriptional burst) likely represent instructive elements for the male germ cell state, hence preferential routes to tamper with this cellular identity.

Through differential gene expression analysis, we identified, in human and mouse spermatocytes, all expressed genes that shared a similar upregulation at meiotic entry and/or downregulation at meiotic exit. Of these 970 mammalian differentially expressed genes (DEGs), we selected as candidates for functional assessment only those that were also expressed in fruit fly germ cells (*Figure 2f*). We did not take into account differential expression in the latter species, since the largely residual levels of post-meiotic transcription in fruit fly spermatogenesis thwarts direct comparisons with the mammalian system (*Lim et al., 2012*). The resulting 920 fruit fly genes (homologous to 797 and 850 in humans and mice, respectively) were silenced specifically at the mitosis-to-meiosis transition by *Drosophila* in vivo RNAi, using the well-established *bam*-GAL4 driver (*White-Cooper, 2012*). Of these, a total of 250 (27.2%) were essential for male fertility, as their silencing resulted, upon mating with wildtype females, in egg hatching rates below the cut-off of 75% (>2 standard deviations of the mean observed in negative controls: 91.6 ± 8.5%; *Figure 2g*). We observed that 76.0% of the hits (190 out of 250) were part of the orthoBackbone, compared with 60.0% (402 out of 670) in genes without any obvious effect on male reproductive fitness upon silencing, with this difference being statistically significant (two-sided Fisher's exact test p<0.0001). Cytological analysis of all 250 genes by testicular phase-contrast microscopy in the RNAi males revealed diverse origins for the infertility phenotype, with the earliest manifestation of cellular defects ranging from the pre-meiotic to the mature gamete stage (*Figure 2h*). By exploring publicly available information (see Materials and methods), we determined that 161 (64.4%) of all hits had never been previously associated with male reproduction in any species (*Figure 2i*). Accordingly, these 161 new deeply conserved spermatogenesis genes (homologous to 179 and 187 in humans and mice, respectively) represent both a significant advance in our understanding of the genetic basis of male germ cell development, and a valuable resource to explore from a precision medicine perspective. To facilitate open access to this information, we made all data available in the form of a user-friendly gene browser (https://pages.igc.pt/meionav; *Figure 2j*). Overall, by merging our results with previously published data we conclude that at least 43.5% of the mitosis-to-meiosis transcriptional burst (our 250 hits +150 previously reported male fertility genes that gave a negative result in our assay) is required for a diverse range of germ cell functions. This illustrates the pervasive influence of the mitosis-to-meiosis transition on multiple facets of the spermatogenic program and argues for a broad functional basis underlying male germ cell identity.

## An ancient regulator of human spermatogenesis

One of our 161 newly identified spermatogenesis genes – the *Drosophila* RING finger protein 113 (*dRNF113*), a spliceosomal component also known as *mdlc* (*Carney et al., 2013*) – emerged as a particularly interesting germ cell regulator from a clinical perspective. More specifically, by analyzing our in-house whole exome database containing sequencing data from 74 cases of human male meiotic arrest, we identified an infertile man harboring a homozygous loss of function (LoF) variant in a human paralog (*RNF113B*). This frameshift variant c.556_565del;p.(Thr186GlyfsTer119) leads to the abrogation of the protein's two functional domains (a C3H1-type zinc finger and a RING finger) and to its truncation (*Figure 3a* and *Figure 3—figure supplement 1a*). The identified man (M1911) is part of a consanguineous family of Middle-Eastern ancestry and shared the homozygous state of the *RNF113B* LoF variant with his equally infertile brother, but not with his fertile heterozygous brother (*Figure 3—figure supplement 2a–c* and Materials and methods). We excluded the possibility that this

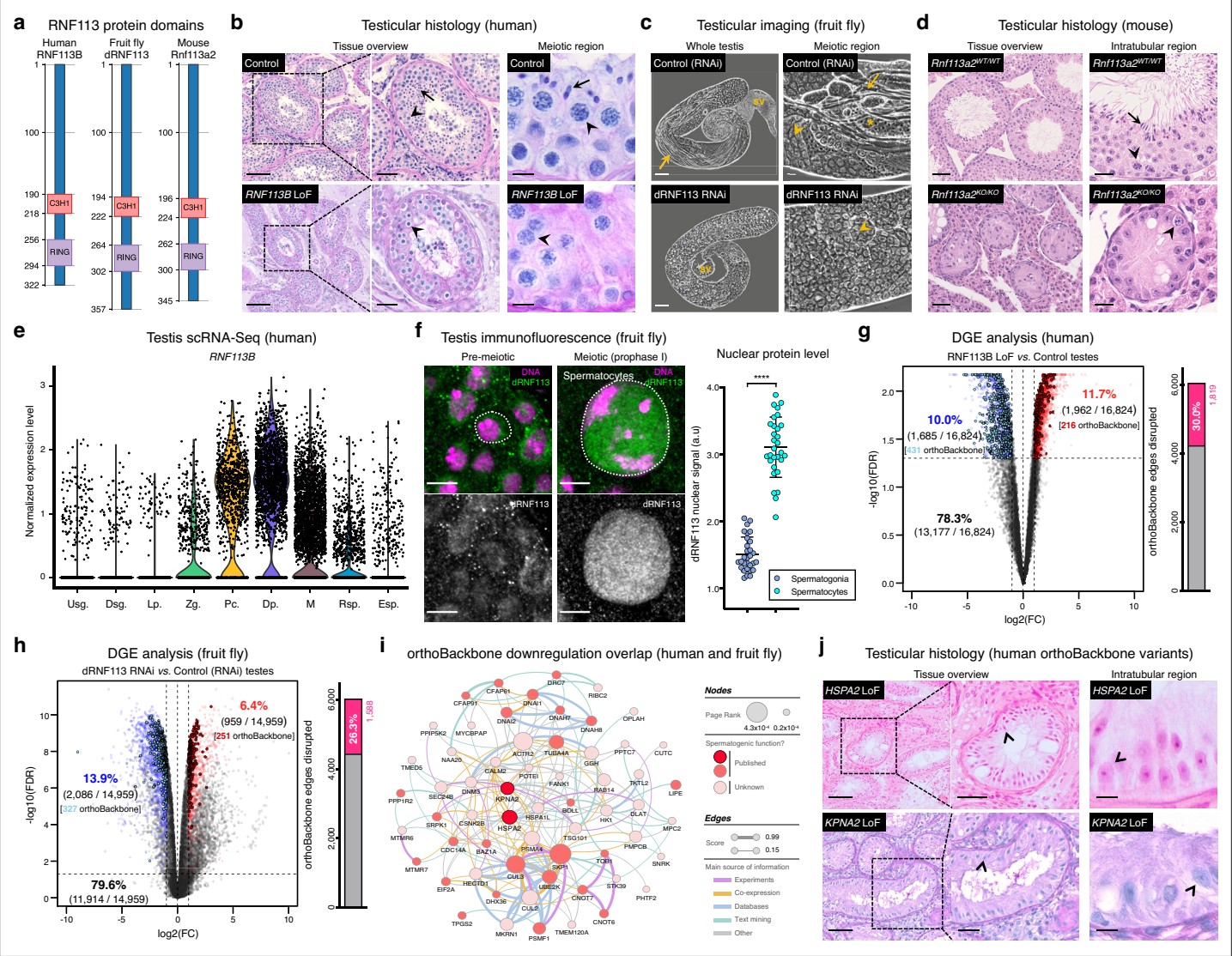

**Figure 3.** Deeply conserved regulators of human spermatogenesis. (**a**) Similar domain structure of the RNF113 proteins in humans (RNF113B), fruit flies (dRNF113), and mice (Rnf113a2). All contain a C3H1-type zinc finger and a RING finger domain. Numbers indicate amino acid residue position. (**b**) Human *RNF113B* is required for meiotic progression past the primary spermatocyte stage. Testicular histology of M1911 [*RNF113B* loss of function (LoF) variant] and of a control sample with normal spermatogenesis. See *Figure 3—figure supplement 1b* for phenotype quantification. Arrowheads: primary spermatocytes; arrows: spermatids. Scale bars: 100 μm (overview), 50 μm (insets), and 10 μm (meiotic region). (**c**) Silencing fruit fly *dRNF113* results in meiotic arrest. Phase-contrast microscopy. See *Figure 3—figure supplement 1c* for phenotype quantification. Arrowheads: primary spermatocytes; asterisks: early (round) spermatids; arrows: late (elongating) spermatids; sv- seminal vesicle. Scale bars: 50 μm (whole testis) and 20 μm (meiotic region). (**d**) Mouse *Rnf113a2* is essential for spermatogenesis. Testes of whole-body homozygous *Rnf113a2* knockout mice (*Rnf113a2^{KO/KO}*) are essentially devoid of germ cells, with the rare occurrence of meiotic and pre-meiotic stages. The testicular histology of a wildtype littermate control (*Rnf113a2^{WT/WT}*) is presented for comparison (normal spermatogenesis). See *Figure 3—figure supplement 1d–g* for additional data. Arrowheads: primary spermatocytes; arrows: spermatids. Scale bars: 50 μm (overview) and 20 μm (intratubular region). (**e**) Human *RNF113B* is predominantly expressed at meiotic entry. Data analyzed from our recently published single cell RNA-Seq atlas of normal spermatogenesis (*Di Persio et al., 2021*). Usg.: undifferentiated spermatogonia; Dsg.: differentiated spermatogonia / preleptotene; Lp.: leptotene; Zg.: zygotene; Pc.: pachytene; Dp.: diplotene; M: meiotic divisions; Rsp.: round spermatids; Esp.: elongating spermatids. (**f**) The nuclear levels of the fruit fly dRNF113 protein increase at meiotic entry. Images are maximum projections of the entire nuclear volume. Spermatocytes correspond to late prophase I cells. Dotted lines delimit the nuclear envelope (as assessed by fluorescent wheat germ agglutinin). Scale bar: 5 μm. a.u- arbitrary units. ****p<0.0001 (unpaired t-test). (**g**) *RNF113B* is required for normal gene expression during human spermatogenesis. Differential gene expression (DGE) analysis of RNA-Seq data obtained from testicular biopsies of M1911 (*RNF113B* LoF variant, left and right testis) and of three controls with normal spermatogenesis. Down and upregulated genes in blue and red, respectively. orthoBackbone differentially expressed genes (DEGs) are outlined. FC: fold change. FDR: false discovery rate. Edge disruption corresponds to the number of orthoBackbone edges containing at least one DEG. (**h**) *dRNF113* regulates gene expression in the fruit fly male gonad. Whole testes

*Figure 3 continued on next page*

*Figure 3 continued*

samples (in triplicate) in both experimental conditions. (**i**) Network of functional associations between orthoBackbone genes downregulated both in the *RNF113B* LoF and dRNF113 RNAi. Node size indicates the result of the page rank metric in the spermatocyte protein-protein interaction network (measure of the connectivity of interacting genes), and color specifies if the gene has a known role in spermatogenesis (in any species). Testicular phenotype of men affected by variants in *HSPA2* and *KPNA2* (red nodes) are depicted in "j". Edge thickness indicates STRING functional association scores, and color specifies the main source of data for the associations. (**j**) LoF variants in the orthoBackbone genes *HSPA2* and *KPNA2* are potentially associated with human male infertility. Testicular histology of individuals M2190 and M2098 (*HSPA2* and *KPNA2* variants, respectively) reveals a complete loss of germ cells (Sertoli cell-only phenotype). Arrowheads: Sertoli cells. Scale bars: 100 µm (overview), 50 µm (insets), and 10 µm (intratubular region).

The online version of this article includes the following figure supplement(s) for figure 3:

**Figure supplement 1.** The RNF113 proteins are required for male germ cell development across vertebrate and invertebrate species.

**Figure supplement 2.** The loss of function variant in *RNF113B* detected in individual M1911.

**Figure supplement 3.** Silencing the spliceosome component *Prp19* in the fruit fly testis.

**Figure supplement 4.** The loss of RNF113B has a considerable impact on the conserved genetic program of spermatogenesis.

**Figure supplement 5.** Association of *HSPA2* loss-of-function variants and the fruit fly Hsc70-1 RNAi with male infertility.

**Figure supplement 6.** Association of *KPNA2* loss-of-function variants and the fruit fly dKPNA2 RNAi with male infertility.

genetic variant is an indicator of another, nearby variant that is homozygous in the affected individuals and represents the actual cause of infertility in both men (Materials and methods). Remarkably, the results from the testicular biopsy of M1911 revealed an equivalent meiotic arrest phenotype to that observed in *dRNF113*-silenced fruit flies. Indeed, spermatocytes were, by far, the predominant cell type in the male gonads of both species: they were the most advanced cell stage observed in 89.0% of all assessed human seminiferous tubules (*vs.* 9% in controls) and occupied an average of 64.1% of the entire testis area in fruit flies (*vs.* 12.3% in controls; ***Figure 3b–c*** and ***Figure 3—figure supplement 1b–c***). Early (round) spermatids were practically absent in both species, with cellular debris accumulating in the post-meiotic region of the fruit fly gonad. The similar reproductive impairment phenotypes recorded in both species highlights the often-complex relationship between tissue specificity and functional significance. In insects, *dRNF113* is a broadly expressed gene essential for organismal viability (***Carney et al., 2013***), whose particular spermatogenic function was uncovered by germ cell-specific silencing. Both roles were functionally uncoupled in primates via the existence of two *RNF113* copies, with the correctly spliced form of *RNF113B* being mainly testis-specific (***Szcześniak et al., 2011***). The existence of an independent duplication event in the rodent lineage offered us the opportunity to further explore this concept. Compared with its human ortholog (*RNF113B*), the mouse duplication (*Rnf113a2*) is broadly expressed, similarly to *Rnf113a1*, human *RNF113A* and fruit fly *dRNF113*. Yet, the generation of a whole-body *Rnf113a2* knockout (KO) mouse revealed a comparable phenotype to that of the human *RNF113B* LoF variant: severe spermatogenic impairment without any overt somatic defects. When compared to wildtype littermates, male homozygous *Rnf113a2*[KO/KO] mice were sterile and had visibly smaller testes that were largely devoid of germ cells (SCO: Sertoli cell-only phenotype) except for the rare presence of spermatogonia and spermatocytes (***Figure 3d*** and ***Figure 3—figure supplement 1d–f***).

Strengthening our interest in the *RNF113* genes was their possible involvement with male germ cell identity. By analyzing our recently published single-cell RNA-Seq dataset of normal human spermatogenesis (***Di Persio et al., 2021***), we observed that *RNF113B* was predominantly expressed at the mitosis-to-meiosis transcriptional burst, peaking at the diplotene stage of prophase I (***Figure 3e***). This mirrored the protein localization pattern of fruit fly dRNF113, characterized by a substantial nuclear accumulation in primary spermatocytes (***Figure 3f***). Also in mice, *Rnf113a2* was expressed at the transcriptional burst, with an earlier expression in undifferentiated spermatogonia being the likely cause of the slightly more severe histological phenotype of the *Rnf113a2*[KO/KO] mice (***Figure 3—figure supplement 1g***). Consistent with their role in gene expression, the analysis of M1911's testicular transcriptome via RNA-Seq revealed the deregulation of 21.7% of all expressed genes (***Figure 3g***). Notably, the deregulation detected in M1911's testicular transcriptome had a clear impact on the spermatocyte orthoBackbone, with 30.0% of its network edges being disrupted (i.e. containing at least one deregulated gene). A similar effect was patent in the dRNF113 RNAi, with 20.3% of the fruit fly testicular transcriptome being deregulated, resulting in the disruption of 26.3% of the orthoBackbone (***Figure 3h***). For comparison, the silencing of *Prp19*, another transcriptional burst gene and

spliceosomal component that RNF113 proteins associate to *Gatti da Silva et al., 2019*, had a lower effect on the orthoBackbone with 15.2% of disrupted edges (significantly lower than in the dRNF113 RNAi; two-sided Fisher's exact test *P*<0.0001), despite a similar meiotic arrest phenotype (*Figure 3—figure supplement 3a-d*). Based on the above, we conclude that the *RNF113* genes have retained a key role in spermatogenesis for more than 600 million years of evolution. Furthermore, they provided us the means to interfere with a sizable fraction of the male germ cell orthoBackbone in humans and fruit flies – an advantage we next explored from a clinical perspective.

## The orthoBackbone as an ancillary tool in clinical genetics

Male infertility is a complex human disease with a poorly defined genetic component. This contributes to a low causative diagnostic yield (typically just 30%), and to a paucity of clinically validated genes (currently just 104, in contrast with the more than 700 already associated with other clinical disorders such as intellectual disability [*Houston et al., 2021*; *Vissers et al., 2016*]). Since male infertility affects up to 12% of all men (*Agarwal et al., 2015*), addressing such knowledge gap is an issue of clear medical importance. To narrow the gap, we explored the possibility that spermatogenesis is particularly sensitive to disturbances in the male germ cell orthoBackbone. Thus, we harnessed the sizeable effect of RNF113B on this network as a means of identifying additional genetic causes of human infertility. Indeed, we noted that 38.0% (30 out of 79) of the functional interactions that define our previously identified core component of the conserved genetic program of spermatogenesis were disrupted in M1911 (i.e. these interactions contained at least one differentially expressed gene; *Figure 3—figure supplement 4*). In particular, we observed that 19.2% (20 out of 104) of this gene subset was found to be downregulated, compared with 10.0% (1685 out of 16,824) of downregulated genes in the testicular transcriptome of M1911, with this difference being statistically significant (two-sided Fisher's exact test *P*=0.0046; *Figure 3g*). Therefore, we posited that the similar spermatogenic impairment recorded in the human *RNF113B* LoF variant and on the fruit fly dRNF113 RNAi ultimately reflected the downregulation, in both species, of a common set of orthoBackbone genes.

By defining the overlap between differentially expressed orthoBackbone genes in the testicular transcriptome of M1911 and of the dRNF113 RNAi, we identified 61 conserved human genes that were similarly downregulated in both species. These formed a connected network (based on STRING data) suggestive of their functional co-involvement in related biological processes (*Figure 3i*). Of the 61 genes, 27 had already been linked to male germ cell defects in different species: four were clinically validated male infertility genes (*CDC14A*, *CFAP91*, *DNAI1* and *DNAI2*), and the other 23 had been previously reported in animal models (see Materials and methods). The fact that among the latter was *BOLL*, one of the oldest known metazoan gametogenesis genes (*Shah et al., 2010*), was a particularly noteworthy observation. We next tested if these 27 genes could be used to identify new genetic causes of human infertility. For this, we analyzed whole-exome sequencing data of 1021 azoospermic men from the MERGE cohort (*Wyrwoll et al., 2020*). Filtering these exomes for LoF variants in the 27 selected orthoBackbone genes revealed two new human male infertility candidate genes: *HSPA2* and *KPNA2*.

*HSPA2* – heat shock protein family A member 2 – is a molecular chaperone of the highly conserved 70 kDa heat shock protein (HSP70) gene family. HSP70 members are involved in cellular proteostasis from bacteria to human, with the mouse and fruit fly *HSPA2* homologs (*Hspa2* and *Hsc70-1* to *Hsc70-5*, respectively) being required for meiotic progression past the primary spermatocyte stage (*Eddy, 1999*; *Azuma et al., 2021*). Indeed, we observed that the silencing of *Hsc70-1* in the fruit fly testis, also with the *bam*-GAL4 driver, resulted in an equivalent meiotic arrest to that of *dRNF113*, strongly suggesting that the misexpression of this chaperone is a central element of the latter phenotype (*Figure 3—figure supplement 5a–b*). Two different heterozygous LoF variants were detected in our male infertility cohort, with this gene having a predicted autosomal-dominant inheritance (*Figure 3—figure supplement 5c–f* and Materials and methods). The first variant, detected in individual M1678, is the early stop-gain variant c.175C>T;p.(Gln59Ter) that truncates more than 90% of the protein, likely leading to nonsense-mediated decay. There is only one individual listed in the gnomAD database with this variant (minor allele frequency = 0.0004%). The second, identified in individual M2190, is the frameshift variant c.1806dup;p.(Glu603ArgfsTer81) which affects the distal end of the protein's nucleotide-binding domain. This variant is not listed in the gnomAD database. Histopathological analysis of M2190's testicular tissue revealed a SCO phenotype in 259 assessed seminiferous tubules, a

likely aggravation of the extensive cell death accompanied by tubule vacuolization reported in *Hspa2* mutant mice (*Dix et al., 1996*; *Figure 3j*).

*KPNA2* – karyopherin subunit alpha 2 – is a nuclear importin that functions as an adapter protein in nucleocytoplasmic transport. Its mouse ortholog (*Kpna2*) is required for the nuclear accumulation of Hop2 (*Ly-Huynh et al., 2011*), a conserved regulator of meiotic progression from yeast to mammals (*Leu et al., 1998*; *Petukhova et al., 2003*). The main function of Hop2 is to repair meiotic DNA double-strand breaks (DSBs), with the corresponding mouse mutant being characterized by a primary spermatocyte arrest coupled to extensive cell death. Since male fruit flies dispense with meiotic DSBs (*John et al., 2016*), the inclusion of *KPNA2* in the male germ cell orthoBackbone seems counter-intuitive. Yet, *Drosophila* importin alpha 2 (*dKPNA2*, also known as *Pendulin*) is required for post-meiotic development (*Mason et al., 2002*), with its silencing specifically at meiotic entry being associated with aborted spermiogenesis (*Figure 3—figure supplement 6a–b*). This post-meiotic function might also be present in its mammalian orthologs, as suggested by the nuclear localization of Kpna2 in elongating mouse spermatids (*Ly-Huynh et al., 2011*). We detected two heterozygous *KPNA2* LoF variants in our cohort of infertile men, with this gene also having a predicted autosomal-dominant inheritance (*Figure 3—figure supplement 6c–g* and Materials and methods). The splice-site variant (c.667–2A>G;p.?) in individual M1645 is predicted to disrupt the correct splicing of intron 6 due to the loss of an acceptor site. Segregation analysis revealed the de novo occurrence of this variant, strongly supporting both its pathogenicity and our recent results, pointing to the important contribution of heterozygous de novo mutations to male infertility (*Oud et al., 2022*; *Wyrwoll et al., 2023*). The other *KPNA2* LoF variant, in individual M2098, is the frameshift variant c.1158_1161del;p.(Ser387ArgfsTer14) which affects 27% of the protein, including its minor nuclear localization signal binding site. Both *KPNA2* variants are not listed in the gnomAD database. The available testicular histopathology report for the latter individual lists a SCO phenotype as the cause of the azoospermia, again suggestive of the possible deterioration of an initially meiotic phenotype (*Figure 3j*).

Based on three in silico prediction tools (Materials and methods), both *HSPA2* and *KPNA2* can represent haploinsufficient genes and/or have autosomal dominant inheritance, with the evidence being stronger for *KPNA2*. Together with the identified de novo occurrence of one of its LoF variants, this makes *KPNA2* a strong novel autosomal-dominant candidate gene while more data is still needed before drawing similar conclusions on *HSPA2*.

In summary, by experimentally disrupting the male germ cell orthoBackbone across evolutionarily distant species, we were able to uncover two new candidate genes for human infertility (*HSPA2* and *KPNA2*) in addition to *RNF113B*, affecting a combined total of five individuals. This successful merger between basic and clinical research highlights the advantages of comparative biology as means of dampening inter-species differences in reproductive physiology, while providing a conceptual framework for a more efficient prioritization of clinically relevant genetic variants in human reproductive disease.

## Discussion

In summary, we have identified a conserved male germ cell gene expression program, spanning more than 600 million years of evolution, that can provide clinical insight into the molecular basis of human infertility. We estimate the average functional requirement of a metazoan male germ cell to correspond to the expression of approximately 10,000 protein-coding genes, the majority of which are deeply conserved. Indeed, despite the testis being a rapidly evolving organ and frequent birthplace of genetic novelty, deeply conserved genes are central components of the male germ cell transcriptome, with younger genes being preferentially expressed in late spermatogenesis. We further pinpoint 79 functional interactions between 104 transcriptional regulators that, owing to their salient network properties across different species, likely define a core component of the ancient genetic program of spermatogenesis. By functionally interfering with 920 deeply conserved genes associated with male germ cell identity, we uncover 161 previously unknown spermatogenesis genes required for a broad range of germ cell functions, as well as three new potential genetic causes of human infertility.

We propose that the transcriptional identity of metazoan male germ cells is built around a relatively small network of deeply conserved gene interactions with an overarching functional impact. These observations may provide a unifying genetic basis for the deep conservation of fundamental germ cell biological processes such as sperm motility and gamete fusion (*Speer et al., 2021*; *Moi et al.,*

*2022*). The existence of a shared genetic identity in metazoan spermatocytes can be regarded as a ramification of an ancestral multipotency program already present in germ line precursor cells (*Fierro-Constaín et al., 2017*). Indeed, even in species with divergent germ line segregation strategies, conserved functional interactions at the post-translational level are required for primordial germ cell specification (*Perillo et al., 2022*; *Colonnetta et al., 2022*). It should be noted that our conclusions were based on a small fraction of the diversity of all metazoan species, as a result of data availability limitations and our focus on well-characterized gonochoric animal models of human spermatogenic impairment. The latter led to the exclusion of established animal models such as *C. elegans* (hermaphrodites) and zebrafish (juvenile hermaphrodites) since it is largely unknown to what extent their male germ cell transcriptome may deviate from the gonochoric program (by retaining oogenesis-related characteristics, for example). The advent of more comprehensive gene expression datasets from less-studied organisms will surely provide a more encompassing context to cross-species comparisons of this nature.

The actual benefit of comparative biology for the identification of new genetic causes of human disease is often a contentious topic (*Pound and Ritskes-Hoitinga, 2018*). Central to this debate is the moderate success in translating animal data to the clinical setting, coupled with the fact that the vast majority of human genetic variants are not shared with other species (*Benton et al., 2021*). By focusing on the deep evolutionary past of human spermatogenesis, we have identified 79 functional interactions that likely represent a core component of the conserved genetic program of male germ cells, and 179 novel functionally validated candidate genes. Exploring to what extent these core functional interactions reflect direct protein-protein contacts is an important area for future research. In this regard, the recent development of robust in silico predictive systems, such as AlphaFold-Multimer, will greatly facilitate the implementation of such studies (*Evans et al., 2021*).

The translational application of our cross-species platform in human reproductive healthcare has so far uncovered three new genes associated with human male infertility (*RNF113B, HSPA2,* and *KPNA2*). The assumed autosomal dominant inheritance of *HSPA2* and *KPNA2* awaits further confirmation by additional data from independent cohorts and segregation analyses, but stands in line with other well-established male infertility genes such as *DMRT1, NR5A1,* and *SYCP2* (*Houston et al., 2021*). Also, this information can potentially be harnessed in other clinically relevant contexts: the spermatocyte orthoBackbone can be further explored to pinpoint key pathways that may facilitate the reconstitution of meiotic entry in vitro, as well as to maximize the effectiveness of exome sequencing approaches in infertile men through the inclusion of the orthoBackbone genes in prioritization lists for disease-causing variants. Conversely, future iterations of our analytical pipeline can be specifically tailored for the identification of newly emerged genes that may have acquired key roles in reproductive processes, such as those involved in species-specific gamete recognition (*Raj et al., 2017*).

Taken together, our results emphasize the often-overlooked contribution of evolutionary history to human disease and illustrate how interdisciplinary research can significantly expand our knowledge of fundamental cellular processes. Accordingly, all the code required for repurposing our analytical pipeline to other cell types and pathologies is available as an open-access resource at https://github.com/rionbr/meionav (copy archived at *Brattig Correia, 2024*). These resources will likely contribute to a renewed appreciation of comparative biology in the medical field.

## Materials and methods
### Phylostratigraphic analysis
For humans, mice and fruit flies, published RNA-Seq data of different stages of spermatogenesis and of representative cell types of the three primary embryonic layers (*Supplementary file 1*) were downloaded from the Sequence Read Archive. Data were checked for quality control and preprocessed by trimming adaptor sequences (Trim galore!; v0.5.0; Babraham Bioinformatics). Gene expression was quantified as transcripts per million (TPM) using Salmon v0.14.1 (*Patro et al., 2017*). Transcript-level information was aggregated at the gene level, thus minimizing the potential issue of transcript diversity. Orthogroups (set of homologous genes derived from the last common ancestor) were defined based on the proteomes of 25 species representing key phylogenetic positions (*Figure 1—figure supplement 1*), and were assembled using OrthoFinder v2.4.0 (*Emms and Kelly, 2019*), with DIAMOND v0.9.24.125 as aligner under default settings (*Buchfink et al., 2015*). A species tree reflecting the

current consensus of the eukaryotic phylogeny (*Dunn et al., 2008*; *Fairclough et al., 2013*; *Laumer et al., 2019*) served as the basis for the phylostratigraphic analysis (*Figure 1—figure supplement 1*). Each orthogroup was assigned to a phylostratum (node) by identifying the oldest clade found in the orthogroup (*Domazet-Loso et al., 2007*), using ETE v3.0 with the 'get_common_ancestor' option. Phylostrata were assigned a node number, ranging from 1 (oldest) to 16 (youngest, species-specific). Only genes with a minimum average TPM >1 were considered expressed. After mapping all expressed genes to the phylostratum containing their corresponding orthogroup, the distribution of the transcriptome allocated to phylostrata 1–5 (orthogroups common to all metazoa) was compared, in a pair-wise manner, between germ cells and soma using the Mann–Whitney U test as implemented in the SciPy python package (*Virtanen et al., 2020*). The transcriptome age index (TAI) of the germ cell samples was calculated by dividing the product of each gene's TPM value and node number by the sum of all TPM values (*Domazet-Lošo and Tautz, 2010*). Higher TAI values represent younger transcriptomes.

## Protein-protein interaction (PPI) network construction

To assemble the spermatocyte PPI networks based on transcriptome data, previously published RNA-Seq datasets of purified human and mouse spermatocytes and of the spermatocyte-enriched median region of the fruit fly testes were used to identify all expressed genes (see *Supplementary file 1* for the list of all selected datasets). Salmon v0.14.1 was used to quantify gene expression levels, and genes were considered expressed based on a minimum average expression level of TPM >1. All potential interactions between expressed genes were retrieved from the STRING v11 database (*Szklarczyk et al., 2019*). For this, we used the combined confidence score, which corresponds to the estimated likelihood of a given association being true, given the underlying evidence. To remove low-confidence interactions, only PPIs with a STRING combined confidence score ≥0.5 were included (*Figure 1—figure supplement 3a–c*). A similar approach was employed to assemble enterocyte PPI networks. A multilayer network representation was employed to define orthologous relationships, with each layer corresponding to a particular species and nodes belonging to the same orthogroup being connected across layers. Orthologous connections were established using the EggNOG v5 database at the metazoan level (*Huerta-Cepas et al., 2019*) and a one-to-many relationship was employed when identifying orthologs between network layers. A variable percentage of all spermatocyte network edges was randomly shuffled to test for possible ascertainment bias in conserved genes (*Figure 1—figure supplement 6*).

## Backbone and orthoBackbone computation

The initial step to define the male germ cell orthoBackbone was the extraction of the metric backbone of each species' spermatocyte PPI network (i.e. each layer of the multilayer network). The metric backbone is the subgraph that is sufficient to compute all shortest paths in the network, thus removing edges that break the triangle inequality (and are therefore redundant in regards to the shortest paths). Importantly, the metric backbone retains all metric edges while preserving all network nodes, the natural hierarchy of connections and community structures (*Simas et al., 2021*; *Brattig Correia et al., 2023*). Considering that a set of the PPIs in each layer are evolutionarily conserved, we computed the orthoBackbone: a single shared subnetwork obtained by collapsing the metric backbone of all layers according to the orthologous relationships. More precisely, the orthoBackbone corresponds to a subgraph of the metric backbone of every layer, where each edge has an analogous edge connecting the same orthologous genes in all other species' layers. For cases where there was not a one-to-one conserved edge relationship (i.e. an edge of one layer's backbone mapped to several edges in other species' layers due to the existence of paralogs), the inclusion criterion was for at least one of these homologous edges to be part of the backbone (*Figure 2a*). Note that because only edges between orthologous genes can be part of the orthoBackbone, nodes (including orthologous nodes) that are left with no edges are removed. Thus, the orthoBackbone typically has fewer nodes than its original network layer and it is not necessarily a metric backbone. Next, annotations retrieved from the Gene Ontology (GO) resource database (release 2020-09-10) were used to select all human orthoBackbone edges connecting at least one spermatogenesis-related gene linked to gene expression regulation. This led to the identification of the 79 functional interactions in spermatocytes that form a core component of the conserved genetic program of spermatogenesis (*Figure 2e*). To determine

the impact of using three evolutionarily distant species to define the orthoBackbone, we calculated the size of this subgraph when calculated based on just two species. We observed that the orthoBackbone of a human and mouse spermatocyte contains 19,333 edges (5.5% of all functional interactions in the human spermatocyte network), and that of a human and fruit fly spermatocyte contains 9,285 edges (2.6% of all functional interactions).

## Differential gene expression analysis of the mitosis-to-meiosis transcriptional burst

Reads were aligned to their respective genomes (GRCh38, GRCm38, and BDGP6.22) using HISAT2 v2.1.0 under default parameters (*Kim et al., 2019*). Uniquely mapped read counts were generated using FeatureCounts v1.5.0-p1 and ENSEMBL GTF annotations (*Liao et al., 2014*). Differential gene expression analysis was performed using a likelihood test (*McCarthy et al., 2012*), as implemented in the edgeR package (*Robinson et al., 2010*). Genes that were significantly upregulated at meiotic entry (spermatocyte *vs.* spermatogonia) and/or downregulated at meiotic exit (spermatid *vs.* spermatocyte) were selected for further analysis as part of the mitosis-to-meiosis transcriptional burst that activates the spermatogenic program. Three criteria were employed to define differentially expressed genes (DEGs): false discovery rate ≤0.05; abs[$\log_2$(fold change)]≥1; and average $\log_2$(normalized counts per million)≥1. Subsequently, EggNOG orthogroups defined at the metazoan level (*Huerta-Cepas et al., 2019*) were used to establish which genes had a similar up/downregulation behavior in both species. This list was further trimmed by only retaining those whose fruit fly orthologs were also expressed at average $\log_2$(normalized counts per million)>1 in the corresponding testes, leading to 920 fruit fly genes (homologous to 797 and 850 in humans and mice, respectively).

## *Drosophila* in vivo RNAi

An in vivo UAS-GAL4 system was used to silence the 920 conserved transcriptional burst genes specifically at the mitosis-to-meiosis transition (*Brand and Perrimon, 1993*). Silencing was induced using the *bam*-GAL4 driver (kindly provided by Renate Renkawitz-Pohl, Philipps Universität Marburg, Germany), which promotes high levels of shRNA expression in late spermatogonia and early primary spermatocytes (*White-Cooper, 2012*). UAS-hairpin lines targeting the selected genes were purchased from the Bloomington *Drosophila* Stock Centre (BDSC) and the Vienna *Drosophila* Resource Centre (VDRC). Lines previously associated with a phenotype in the literature, regardless of the tissue, were preferentially chosen for this experiment. In the 10 cases where no lines were available, these were generated in-house following standard procedures (*Perkins et al., 2015*). Briefly, shRNA were designed using DSIR (*Vert et al., 2006*), with the corresponding sequences being cloned into the pWalium 20 vector. Constructs were injected into fruit flies carrying an attp site on the third chromosome (BDSC stock #24749) and transgene-carrying progeny were selected to establish the UAS-hairpin lines. A similar strategy was employed to generate a second independent RNAi reagent for dRNF113 (dRNF113 #2). The antisense sequences selected for each locus are listed in *Supplementary file 3*, and information on all tested lines is available at the Meiotic Navigator gene browser.

For assessing fertility, gene-silenced males were mated with wildtype Oregon-R virgin females (2 males:4 females) for 12 hours at 25 °C. Laid eggs were left to develop for 24 hr at 25 °C before the percentage of egg hatching was determined. This percentage (fertility rate) served as a measure of the male reproductive fitness associated with each tested gene. All fruit flies were 3–7 days post-eclosion and fertility rates correspond to an average of four independent experiments, with a minimum of 25 eggs scored per replicate. Every batch of experiments included a negative control [RNAi against the mCherry fluorophore (BDSC stock #35785) – a sequence absent in the fruit fly genome], and a positive control [RNAi against Ribosomal protein L3 (BDSC stock #36596) – an essential unit of the ribosome]. A cut-off of <75% fertility rate was established to define impaired reproductive fitness based on the rate observed in the negative controls (>2 standard deviations of 91.6 ± 8.5%). For dRNF113, Hsc70-1 (VDRC stock #106510) and dKPNA2 (VDRC stock #34265) extended fertility tests were conducted, with ~100 eggs scored per replicate. To this end, egg-laying cages with apple juice agar plates as substrate were set up with a 1:2 male to female ratio. All *Drosophila* lines were maintained at 25 °C in polypropylene bottles containing enriched medium (cornmeal, molasses, yeast, beet syrup, and soy flour).

The novelty of the hits was determined based on the following exclusion criteria (applied to all 250 hits and their orthologs): i- clinically validated human infertility genes (*Houston et al., 2021*); ii- genes associated with a mouse male infertility phenotype (MP:0001925) in the Mouse Genome Informatics database (*Baldarelli et al., 2021*); and iii- fruit fly genes with an associated male germ cell or male reproductive phenotype in the FlyBase database (*Larkin et al., 2021*). For the latter two categories, information reflects data available in both systems as of April 2023.

## Human and mouse testicular imaging

Testicular biopsies of the azoospermic individuals M1911 (*RNF113B* variant), M2098 (*KPNA2* variant) and M2190 (*HSPA2* variant) were collected at the Centre for Reproductive Medicine and Andrology (CeRA) in Münster, Germany or at the University Hospital Giessen, Germany. As a control for testicular imaging, tissue of individual M2951 (who underwent a vasectomy reversal at the CeRA) was used. All testicular tissues were fixed in Bouin's solution and embedded in paraffin using an automatic ethanol and paraffin row (Bavimed Laborgeräte GmbH). Serial sections (5µm-thick) were stained with periodic acid-Schiff according to standard procedures. Testicular histopathology was performed as part of the routine clinical work up. For quantification, at least 100 testicular tubules per testis were screened for the most advanced germ cell type present (categories: spermatogonia, spermatocytes, round spermatids and elongating spermatids), or in their absence, for Sertoli cells or tubular shadows.

Mouse testes were dissected from 10 week-old animals and fixed in modified Davidson's fluid for 48 hr at room temperature before being paraffin-embedded according to standard procedures (*Meistrich and Hess, 2013*). Serial sections (3µm-thick) were stained with haematoxylin-eosin. Slides were analyzed with the NanoZoomer-SQ Digital slide scanner (Hamamatsu Photonics).

## *Drosophila* testicular imaging

Squash preparations of freshly dissected *Drosophila* testes were performed as previously described (*Bonaccorsi et al., 2011*) and examined using a phase contrast microscope (Nikon Eclipse E400). Phenotypes of all silenced genes associated with decreased reproductive fitness (250 in total) were assigned to one of five classes, based on the earliest stage in which the cytological defects were detected: 1- pre-meiotic (evidence of increased cell death and/or clear morphological defects in late spermatogonia); 2- meiotic (failure to progress successfully through meiosis, evidence of increased cell death, and/or clear morphological defects in spermatocytes); 3- post-meiotic (abnormal spermiogenesis characterized by sperm individualization defects and/or significant spermatid morphological defects); 4- gamete (decreased mature sperm numbers and/or motility compared with controls after qualitative assessment); 5- undetectable (no cytological defects). Images from at least 2 pairs of testes were acquired per genotype and were corrected for background illumination as previously described (*Landini, 2020*). These are all available in the Meiotic Navigator gene browser.

For the quantification of the meiotic area, phase-contrast images of different testicular regions were acquired at 40 x and stitched together, using the MosaicJ tool (*Thévenaz and Unser, 2007*) in the ImageJ software (v1.8, National Institutes of Health), to reconstruct a high-resolution image of the entire testis. On average, eight individual images were acquired to assemble each complete testis. The meiotic area corresponds to the ratio between the area (in pixels) occupied by primary spermatocytes and that of the entire gonad. A total of 15 different testes were quantified per genotype across 3 independent experiments, and groups were compared using unpaired t-tests.

For the dRNF113 immunofluorescence assay, a modified whole-mount protocol specifically designed for the analysis of *Drosophila* spermatocytes was employed (*Raich et al., 2020*). Briefly, testes were dissected in testis buffer (183 mM KCl, 47 mM NaCl, 10 mM Tris-HCl, 1 mM EDTA and 1 mM PMSF), transferred to a pre-fix solution containing 4% formaldehyde (PolySciences) in PBS, and then fixed for 20 min using a heptane-fixative mix at 3:1. The fixative consisted of 4% formaldehyde in a PBS +0.5% NP-40 (Merck) solution. After washing the fixative, samples were incubated for 1 hr in PBS supplemented with 0.3% Triton X-100 (Sigma-Aldrich), 1% (w/v) bovine serum albumin (BLIRT) and 1% (w/v) donkey serum (Thomas Fisher Scientific). Primary antibody incubation (anti-dRNF113 at 1:250; kindly provided by Chris Doe, University of Oregon, USA) was performed overnight at 4 °C in PBS supplemented with 1% (w/v) bovine serum albumin and 1% (w/v) donkey serum. After washing the primary antibody solution, samples were incubated for 1 hr with a goat anti-guinea pig secondary antibody (Alexa Fluor 488 conjugate; 1:1000; Invitrogen) and wheat germ agglutinin to label the

nuclear envelope (Alexa Fluor 647 conjugate; 1:500; Invitrogen). Testes were mounted in Vectashield mounting medium with DAPI (Vector Laboratories) and the entire nuclear volume of individual cells was acquired as 1µm-thick slices using a 63 x oil immersion objective and 10 x digital zoom in a Leica SP5 confocal microscope. Slices were stacked into maximum intensity Z-projections and the relative intensity of the dRNF133 signal was measured as previously described (*Prudêncio et al., 2018*). A total of 30 late spermatogonia and 30 mature primary spermatocytes (late prophase I) were quantified across three independent experiments, and the two groups were compared using an unpaired t-test.

## Generation of genetically modified mice

A mutant allele for the mouse *Rnf113a2* gene was generated using the CRISPR/Cas9 system in the FVB/N background. gRNA was assembled by in vitro hybridization of an Alt-R scRNA targeting the sequence ATGATCCAGAAGACGAGTGG with the Alt-R tracrRNA (both from Integrated DNA Technologies, Inc). An active ribonucleoprotein (RNP) complex was obtained by incubation of the gRNA with the Cas9 protein. The RNP containing 1 µM of the hybridized gRNA and 100 ng/µl of Cas9 were microinjected into the pronucleus of fertilized oocytes and transferred into pseudopregnant females according to standard procedures (*Hogan et al., 1994*). The resulting pups were genotyped by PCR using primers Rnf113-F and Rnf113-R (*Supplementary file 3*). PCR products were run on a native 15% polyacrylamide gel in TBE buffer, from which 3 pups were identified with a PCR pattern consisting of a mix between a fragment matching the expected size for the wildtype allele and a smaller fragment. The smaller fragments were recovered, amplified, and sequenced, revealing deletions of 3, 9, or 14 nucleotides. Only the allele containing the 14-nucleotide deletion is expected to disrupt *Rnf113a2* by generating a frameshift leading to an immediate stop codon after amino acid 64 of the 338 in the full-length. A mouse line was generated from this founder protein (*Rnf113a2*$^{em1Mllo}$, here forth referred to as *Rnf113a2*$^{KO}$). Homozygous *Rnf113a2*$^{KO/KO}$ mice were generated by intercrosses between heterozygous mice. Genotyping was performed by PCR using primers Rnf113-F and Rnf113-Chk-wt-R for the wildtype allele and Rnf113-F and Rnf113-Chk-mut-R for the mutant allele (*Supplementary file 3*). All animals were housed at the Mouse Facility of Instituto Gulbenkian de Ciência (MF-IGC). Wildtype FVB/N mice were generated in the production area of the MF-IGC. Transgenic animals were produced and kept in the barrier experimental area of the MF-IGC.

## Characterization of mouse reproductive parameters

Upon sexual maturation, 8-week-old male F2 *Rnf113a2*$^{KO/KO}$ homozygous mice and corresponding wildtype littermates (*Rnf113a2*$^{WT/WT}$) were caged individually and mated with two 8-week-old wildtype FVB/N females. Five males were tested for each genotype to meet the requirements for statistical validity. After the mating period, females with a confirmed copulatory plug were separated. To promote animal welfare and to reduce the number of animals used, we avoided the normal delivery of pups as a means of assessing fertility. For this, separated females were euthanized by cervical dislocation 10 days post copulation and male fertility was evaluated by the presence of developing embryos. Whenever present, the number of embryos was recorded to quantitate the number of progeny sired by the tested males. All males were considered tested after at least two females had confirmed copulatory plug. To complement the in vivo fertility tests, 10-week-old males were euthanized by cervical dislocation for testicular collection and to examine internal anatomy. Testes were weighted before being processed for histology (see 'Human and mouse testicular imaging'). All experiments followed the Portuguese (Decreto-Lei n° 113/2013) and European (Directive 2010/63/EU) legislations, concerning housing, husbandry, and animal welfare.

## RNA-Seq in human and *Drosophila* testes

Total RNA from two snap-frozen testicular tissue samples of the azoospermic individual M1911 (*RNF113B* variant) as well as from one sample each from three unrelated individuals with normal spermatogenesis (M1544 [obstructive azoospermia], M2224 [anorgasmia] and M2234 [previous vasectomy]; *Siebert-Kuss et al., 2023*), were extracted using the Direct-zol RNA Microprep kit (Zymo Research), following the manufacturer's instructions. RNA quality was estimated by electrophoresis (Agilent Technologies), with all samples having a RNA integrity number (RIN) >4.5 (range: 4.5–5.6), except one replicate of M1911 (with RIN = 2.0). rRNA depletion was performed with the NEBNext rRNA Depletion Kit v2 (New England Biolabs), and total RNA libraries were prepared with the

NEBNext Ultra II Directional RNA Library Prep Kit for Illumina (New England Biolabs), using 700 ng of RNA per sample. Paired-end sequencing with 150 bp per read was performed using the NextSeq2000 system (Illumina) at the University of Münster, Germany. An average of 44 million reads was generated per sample. The expression pattern of *RNF113B* was determined in a control testis by retrieving the corresponding data from our recently published single-cell RNA-Seq dataset of normal human spermatogenesis (*Di Persio et al., 2021*).

For fruit flies, RNA was extracted from 40 pairs of adult testes per sample per condition (3–7 days post-eclosion) using the PureLink RNA Mini Kit (Thermo Fisher Scientific) following the manufacturer's instructions. Three conditions were analyzed: dRNF113 RNAi (meiotic arrest), Prp19 RNAi (meiotic arrest) and the mCherry RNAi negative control (normal spermatogenesis), with three independent biological replicates per condition. Extracted RNA was treated with DNAse (Thermo Fisher Scientific), with a RNA quality number of 10 for all samples, as assessed by electrophoresis (Advanced Analytical Technologies). Total RNA libraries were prepared using the Zymo-Seq RiboFree Total RNA-Seq Library Kit (Zymo Research), using 1000 ng of RNA per sample. Paired-end sequencing with 150 bp per read was performed using DNBSEQ technology (BGI Group), with an average of 46 million reads per sample. Confirmation of knockdown efficiency via quantitative RT-PCR was performed for both the dRNF113 and Prp19 RNAi. For this, cDNA was synthetized using the Transcriptor First Strand cDNA Synthesis Kit (Roche), according to the manufacturer's instructions. Reactions were performed in a QuantStudio 6 Real-Time PCR System (Thermo Fisher Scientific), using SYBR Green chemistry (Applied Biosystems). All reactions were set up in triplicates and Act57B and RpL32 were used as internal controls. Primers are listed in *Supplementary file 3*.

RNA-Seq reads were aligned against the human and fruit fly genomes (GRCh38 and BDGP6.32, respectively) using the STAR aligner (*Dobin et al., 2013*). Gene-level counts were obtained using feature-counts, taking into account the strandedness. Counts were normalized with the TMM method (*Robinson and Oshlack, 2010*), and differential gene expression analysis was performed using a quasi-likelihood F-test (*Lun et al., 2016*), as implemented in the edgeR package. DEGs were defined using the previously listed criteria (see 'Differential gene expression analysis of the mitosis-to-meiosis transcriptional burst'). To determine to what extent the human patient sample with lower RIN could be affecting our results, we repeated this analysis correcting for sample degradation using DegNorm (*Xiong et al., 2019*) This revealed that 97.8% of all DEGs, as defined by our criteria, were also listed as such after taking into account sample degradation. This fits with a largely similar number of expressed genes (at TPM >1) detected in the control and in the experimental groups (average of 53.9% and 46.8% expressed protein-coding genes, respectively).

## Whole exome sequencing and orthoBackbone analysis

Whole exome sequencing (WES) in the 1,021 azoospermic men included in the MERGE study cohort was performed as previously described (*Wyrwoll et al., 2020*). All men provided written informed consent, in agreement with local requirements. The study protocol was approved by the Münster Ethics Committees/Institutional Review Boards (Ref. No. Münster: 2010–578 f-S) in accordance with the Helsinki Declaration of 1975. Exome sequencing data were filtered for high-confidence loss-of-function (LoF) variants in the 27 orthoBackbone genes that were similarly downregulated in the testicular transcriptome of M1911 and of the dRNF113 RNAi, and for which there were available functional data supporting a role in spermatogenesis. These genes were: *BAZ1A, BOLL, CDC14A, CFAP61, CFAP91, CNOT6, CNOT7, CUL3, DHX36, DNAH7, DNAH8, DNAI1, DNAI2, DRC7, EIF2A, HSPA2, KPNA2, LIPE, MTMR7, PPP1R2, PSMF1, SKP1, SRPK1, TOB1, TPGS2, TUBA4A,* and *UBE2K*. From this list, all those previously associated with known causes of human infertility or with congenital syndromic conditions were excluded (*CFAP61, CFAP91, DNAH7, DNAH8, DNAI1, DNAI2,* and *LIPE*). The inheritance mode for each candidate gene was predicted using three sources: the DOMINO algorithm (*Quinodoz et al., 2017*), the gnomAD observed/expected (o/e) constraint score for LoF variants (*Karczewski et al., 2020*), and the recent dosage sensitivity map of the human genome (*Collins et al., 2022*). Filtering criteria were: stop-gain, frameshift, and splice site variants with a minor allele frequency (MAF) <0.01 in gnomAD, a maximum occurrence of 10 x in our in-house database and a read depth >10. Variants were only considered when detected in accordance with the predicted mode of inheritance.

## Clinical genetics

The homozygous frameshift variant c.556_565del;p.(Thr186GlyfsTer119) in *RNF113B* in individual M1911 was identified by filtering WES data of 74 men with complete bilateral meiotic arrest for rare LoF variants. Two first degree cousins of M1911 were also infertile, but genetic analyses were not possible. It is likely that both are also affected by the homozygous *RNF113B* frameshift variant, as the parents of both men are consanguineous (therefore, family ascendants of M1911). The heterozygous LoF variants in the *HSPA2* and *KPNA2* genes (2 cases for each, for a total of four unrelated men) were identified by analyzing data from 1,021 azoospermic men from the MERGE cohort. For *HSPA2*, the heterozygous stop-gain variant c.175C>T;p.(Gln59Ter) was identified in individual M1678, and the heterozygous frameshift variant c.1806dup;p.(Glu603ArgfsTer81) in M2190. For *KPNA2*, the heterozygous splice acceptor variant c.667–2A>G;p.? was detected in individual M1645, and the heterozygous frameshift variant c.1158_1161del;p.(Ser387ArgfsTer14) in M2098.

The inheritance for *HSPA2* and *KPNA2* is predicted to be 'very likely autosomal dominant' by DOMINO. The gnomAD o/e scores are 0.32 (0.17–0.67) for *HSPA2* and 0.21 (0.11–0.44) for *KPNA2*. The pHaplo scores are 0.350 for *HSPA2* and 0.747 for *KPNA2*. The proposed 'hard'/conservative thresholds for the 'loss of function observed/expected upper bound fraction' or 'LOEUF' score (the upper limit of the confidence interval) is <0.35 and for pHaplo >0.86. Thus, both LOEUF and pHaplo hint towards likely autosomal dominant inheritance, more so for *KPNA2* than for *HSPA2*, but the formal cut-offs are not met in both cases.

We tested the possibility that the actual cause of infertility in M1911 and his infertile brother was due to another homozygous variant located in the vicinity of *RNF113B*. To this end, we queried all genes within a range of 1 million base pairs in both directions from *RNF113B*. The following 12 genes were analyzed: *FARP1*, *IPO5*, *RAP2A*, *MBNL2*, *MIR3170*, *STK24*, *STK-24AS1*, *SLC15A1*, *DOCK9-AS1*, *DOCK9*, *LINC00456*, and *DOCK9-DT*. Of these, only *IPO5* is robustly expressed in the testis (data: GTEx Project, NIH). Variants in this gene have been associated with keratoconus, but not with male infertility. Additionally, no coding variants affecting any of these 12 genes (MAF <1%) were detected in M1911. These results render extremely unlikely the possibility that M1911's infertility phenotype is caused by a co-segregating variant.

To rule out possible alternative monogenic causes for the infertility phenotype in all five selected individuals, the sequences of 230 genes previously identified with an at least limited level of evidence of being associated with male infertility (*Houston et al., 2021*), were screened in the exomes of M1911 (*RNF113B* variant), M1678 (*HSPA2* variant), M2190 (*HSPA2* variant), M1645 (*KPNA2* variant), and M2098 (*KPNA2* variant). Of these variants, only those affecting genes with a quantitative impact on spermatogenesis, that were consistent with the reported mode of inheritance of the respective gene, and occurring <10 x in our in-house database, were considered for further analysis. These criteria identified the heterozygous missense variant c.2377G>T;p.(Ala793Ser) in *DNMT1* as a possible alternative cause for male infertility in M1911, a possibility that was disproven by subsequent familial studies. Briefly, DNA from M1911's mother, a fertile brother and an azoospermic brother were available for segregation analysis, which was performed by Sanger sequencing. The azoospermic brother of M1911 was also homozygous for the frameshift variant in *RNF113B*, while the mother and the fertile brother were heterozygous for the variant. Of note, the missense variant c.2377G>T p.(Ala793Ser) in *DNMT1* was also identified in M1911's fertile brother in a similarly heterozygous state (primers are listed in *Supplementary file 3*). The latter result strongly suggests that the *DNMT1* variant has no pathogenic effect on male fertility. Segregation analysis was also performed in M1654's parents, revealing that the splice-site variant c.667–2A>G;p.? was not present either in his father or in his mother (i.e. indicating that it was a de novo variant). Paternity was confirmed by short-tandem-repeat analysis. No chromosomal aberrations or Y-chromosomal microdeletions in the AZF regions were identified in any of the selected individuals.

## Clinical data of individuals M1678 and M2190 (*HSPA2* variants)

The testicular phenotype of M1678 is unknown, as this individual did not undergo a testicular biopsy. However, FSH levels of 26.7 U/L (normal range 1–7 U/L) and a bi-testicular volume of 10 mL (normal value >24 mL) are clearly indicative of non-obstructive azoospermia. M2190 underwent a testicular biopsy at the Urology and Andrology department of the University Hospital of Giessen, Germany. He was diagnosed with complete bilateral SCO and sperm retrieval was unsuccessful.

## Clinical data of individuals M1645 and M2098 (*KPNA2* variants)

Individual M1645 underwent a testicular biopsy in a different clinical unit, with the corresponding pathology report stating complete bilateral SCO without successful sperm retrieval. Histology data was not available for this man. M2098 was also diagnosed with complete bilateral SCO and sperm retrieval was equally unsuccessful.

## Acknowledgements

We gratefully acknowledge all men that gave their informed consent to be included in the Male Reproductive Genomics (MERGE) study cohort. The authors wish to thank Gabriel Martins and José Marques from the Imaging Unit of Instituto Gulbenkian de Ciência (IGC, Portugal) for assistance in acquiring and processing microscopy data. We acknowledge the IGC's Histology Facility and Genomics Unit, respectively, for the analysis of mouse testicular tissue and preparation of the fruit fly RNA-Seq libraries, Nicole Terwort (University of Münster) for preparing the RNA extractions from human testes, and the Core Facility Genomics of the Medical Faculty of the University of Münster for conducting library preparation and RNA sequencing in samples from individual M1911. Renate Renkawitz-Pohl (Philipps-Universität Marburg, Germany) and Chris Doe (University of Oregon, USA) kindly provided us the *bam*-GAL4 line and the anti-dRNF113 antibody, respectively. We thank Élio Sucena, Raquel Oliveira and Mónica Bettencourt-Dias (all from the IGC), Patrícia Beldade (Faculty of Sciences, University of Lisbon), and Peter Ellis (University of Kent, UK) for insightful discussions.

## Additional information

### Competing interests

Paulo Navarro-Costa: The other authors declare that no competing interests exist.

### Funding

| Funder | Grant reference number | Author |
|---|---|---|
| Fundação para a Ciência e a Tecnologia | PTDC/MEC-AND/30221/2017 | Paulo Navarro-Costa |
| Fundação para a Ciência e a Tecnologia | EXPL/MEC-AND/0676/2021 | Paulo Navarro-Costa |
| Fundação para a Ciência e a Tecnologia | CEECIND/03345/2018 | Jörg D Becker |
| Fundação para a Ciência e a Tecnologia | PD/BD/114362/2016 | Chandra Shekhar Misra |
| National Institutes of Health | 1R01LM012832 | Luis M Rocha |
| National Science Foundation | Research Traineeship grant 1735095 | Luis M Rocha |
| Deutsche Forschungsgemeinschaft | 329621271 | Frank Tüttelmann |
| Ministry of Education, Singapore | MOE2018-T2-2-053 | Irene Julca Marek Mutwil |

The funders had no role in study design, data collection and interpretation, or the decision to submit the work for publication.

### Author contributions

Rion Brattig-Correia, Resources, Software, Formal analysis, Investigation, Visualization, Methodology, Conceptualization; Joana M Almeida, Margot Julia Wyrwoll, Ana Nóvoa, Ana S Leocádio, Joana Bom, Formal analysis, Investigation, Methodology; Irene Julca, Formal analysis, Investigation, Methodology, Conceptualization; Daniel Sobral, Formal analysis, Investigation, Visualization, Methodology; Chandra Shekhar Misra, Data curation, Investigation, Methodology, Conceptualization; Sara Di Persio,

Leonardo Gastón Guilgur, Hans-Christian Schuppe, Pedro Prudêncio, Sandra Laurentino, Investigation; Neide Silva, Formal analysis, Investigation; Moises Mallo, Validation, Investigation, Methodology; Sabine Kliesch, Validation; Marek Mutwil, Validation, Investigation, Methodology, Funding acquisition, Writing – review and editing, Conceptualization; Luis M Rocha, Frank Tüttelmann, Conceptualization, Validation, Investigation, Methodology, Funding acquisition, Writing – review and editing; Jörg D Becker, Conceptualization, Supervision, Investigation, Methodology, Funding acquisition, Writing – review and editing; Paulo Navarro-Costa, Conceptualization, Formal analysis, Supervision, Funding acquisition, Validation, Investigation, Visualization, Methodology, Writing – original draft, Writing – review and editing

**Author ORCIDs**
Margot Julia Wyrwoll ⓘ https://orcid.org/0000-0002-7651-2403
Daniel Sobral ⓘ https://orcid.org/0000-0003-3955-0117
Chandra Shekhar Misra ⓘ https://orcid.org/0000-0002-3544-4723
Sara Di Persio ⓘ https://orcid.org/0000-0002-9279-7373
Neide Silva ⓘ https://orcid.org/0000-0002-3154-2959
Ana Nóvoa ⓘ https://orcid.org/0000-0002-5668-5630
Sandra Laurentino ⓘ https://orcid.org/0000-0002-5213-2756
Moises Mallo ⓘ https://orcid.org/0000-0002-9744-0912
Frank Tüttelmann ⓘ https://orcid.org/0000-0003-2745-9965
Jörg D Becker ⓘ http://orcid.org/0000-0002-6845-6122
Paulo Navarro-Costa ⓘ https://orcid.org/0000-0002-8543-4179

**Ethics**
All men provided written informed consent, in agreement with local requirements. The study protocol was approved by the Münster Ethics Committees/Institutional Review Boards (Ref. No. Münster: 2010-578-f-S) in accordance with the Helsinki Declaration of 1975.
All mouse experiments followed the Portuguese (Decreto-Lei no 113/2013) and European (Directive 2010/63/EU) legislations, concerning housing, husbandry, and animal welfare.

Reviewer #1 (Public review): https://doi.org/10.7554/eLife.95774.3.sa1
Reviewer #2 (Public review): https://doi.org/10.7554/eLife.95774.3.sa2
Author response https://doi.org/10.7554/eLife.95774.3.sa3

---

# Additional files

**Supplementary files**
• Supplementary file 1. RNA-Seq datasets.
• Supplementary file 2. orthoBackbone genes.
• Supplementary file 3. Primers list.
• MDAR checklist

**Data availability**
An interactive online application compiling experimental results on 920 evolutionarily conserved spermatocyte genes and associated microscopy images (Meiotic Navigator) is available at https://pages.igc.pt/meionav. Computational data and custom R and Python scripts used in the analysis are available from https://github.com/rionbr/meionav (copy archived at *Brattig Correia, 2024*). All testicular RNA-Seq data generated in this study (*Supplementary file 1*) were deposited in the European Genome-Phenome Archive (human data) and in the Sequence Read Archive (fruit fly data).

The following datasets were generated:

| Author(s) | Year | Dataset title | Dataset URL | Database and Identifier |
|---|---|---|---|---|
| Correia B | 2022 | Control RNAi Replicate1 | https://www.ncbi.nlm.nih.gov/sra/SRX12573744 | NCBI Sequence Read Archive, SRX12573744 |
| Correia B | 2022 | Control RNAi Replicate2 | https://www.ncbi.nlm.nih.gov/sra/SRX12573745 | NCBI Sequence Read Archive, SRX12573745 |
| Correia B | 2022 | Control RNAi Replicate3 | https://www.ncbi.nlm.nih.gov/sra/SRX12573746 | NCBI Sequence Read Archive, SRX12573746 |
| Correia B | 2022 | dRNF113 RNAi Replicate1 | https://www.ncbi.nlm.nih.gov/sra/SRX12573747 | NCBI Sequence Read Archive, SRX12573747 |
| Correia B | 2022 | dRNF113 RNAi Replicate2 | https://www.ncbi.nlm.nih.gov/sra/SRX12573748 | NCBI Sequence Read Archive, SRX12573748 |
| Correia B | 2022 | dRNF113 RNAi Replicate3 | https://www.ncbi.nlm.nih.gov/sra/SRX12573749 | NCBI Sequence Read Archive, SRX12573749 |
| Correia B | 2022 | Prp19 RNAi Replicate1 | https://www.ncbi.nlm.nih.gov/sra/SRX12573750 | NCBI Sequence Read Archive, SRX12573750 |
| Correia B | 2022 | Prp19 RNAi Replicate2 | https://www.ncbi.nlm.nih.gov/sra/SRX12573751 | NCBI Sequence Read Archive, SRX12573751 |
| Correia B | 2022 | Prp19 RNAi Replicate3 | https://www.ncbi.nlm.nih.gov/sra/SRX12573752 | NCBI Sequence Read Archive, SRX12573752 |
| Correia B | 2022 | Total RNA sequencing in M1911 | https://ega-archive.org/datasets/EGAD00001008639 | European Genome-Phenome Archive, EGAD00001008639 |
| Correia B | 2022 | Stage-specific gene and transcript dynamics in human male germ cells | https://ega-archive.org/datasets/EGAD00001008652 | European Genome-Phenome Archive, EGAD00001008652 |

The following previously published datasets were used:

| Author(s) | Year | Dataset title | Dataset URL | Database and Identifier |
|---|---|---|---|---|
| Jan SZ, Volmmer TL, Jongejan A, Roling MD, Silber SJ, de Rooji DG, Hamer G, Repping S, van Pelt AMM | 2016 | *Homo sapiens* male germ cell transcriptome | https://www.ncbi.nlm.nih.gov/bioproject/PRJNA310976 | NCBI BioProject, PRJNA310976 |
| Soumillon M, Necsulea A, Weier M, Brawand D, Zhang X, Gu H, Barthès P, Kokkinaki M, Nef S, Gnirke A, Dym M, de Massy B, Mikkelsen TS, Kaessmann H | 2013 | Cellular source and mechanisms of high transcriptome complexity in the mammalian testis (mouse) | https://www.ncbi.nlm.nih.gov/bioproject/PRJNA187158 | NCBI BioProject, PRJNA187158 |
| Vedelek V, Bodai L, Grézal G, Kovács B, Boros LM, Laurinyecz B, Sinka R | 2018 | Analysis of *Drosophila melanogaster* testis transcriptome | https://www.ncbi.nlm.nih.gov/bioproject/PRJNA496287 | NCBI BioProject, PRJNA496287 |

*Continued on next page*

*Continued*

| Author(s) | Year | Dataset title | Dataset URL | Database and Identifier |
|---|---|---|---|---|
| Pharmaceuticals Regeneron | 2019 | RNA Sequencing Of Intestinal Mucosa in Celiac patients | https://www.ncbi.nlm.nih.gov/bioproject/PRJNA528755 | NCBI BioProject, PRJNA528755 |
| Kazakevych J, Sayols S, Messner B, Krienke C, Soshnikova N | 2016 | Dynamic changes in chromatin states during specification and differentiation of adult intestinal stem cells (mouse) | https://www.ncbi.nlm.nih.gov/bioproject/PRJNA352951 | NCBI BioProject, PRJNA352951 |
| Dutta D, Dobson AJ, Houtz PL, Gläßer C, Revah J, Korzelius J, Patel PH, Edgar B, Buchon N | 2014 | Transcriptome profiling of *Drosophila* intestinal cells | https://www.ncbi.nlm.nih.gov/bioproject/PRJNA260852 | NCBI BioProject, PRJNA260852 |
| Xu X, Stoyanova EI, Lemiesz A, Xing J, Mash DC, Heintz N | 2017 | Species and Cell-Type Properties of Classically Defined Human and Rodent Neurons and Glia (human) | https://www.ncbi.nlm.nih.gov/bioproject/PRJNA395921 | NCBI BioProject, PRJNA395921 |
| Lindholm ME, Huss M, Solnestam BW, Kjellqvist S, Lundeberg J, Sundberg CJ | 2014 | The human skeletal muscle transcriptome - sex differences, alternative splicing and tissue homogeneity assessed with RNA sequencing | https://www.ncbi.nlm.nih.gov/bioproject/PRJNA252429 | NCBI BioProject, PRJNA252429 |
| Lindholm ME, Huss M, Solnestam BW, Kjellqvist S, Lundeberg J, Sundberg CJ | 2017 | PRJNA395915; Species and Cell-Type Properties of Classically Defined Human and Rodent Neurons and Glia (mouse) | https://www.ncbi.nlm.nih.gov/bioproject/PRJNA395915 | NCBI BioProject, PRJNA395915 |
| Terry EE, Zhang X, Hoffmann C, Hughes LD, Lewis SA, Li J, Wallace MJ, Riley LA, Douglas CM, Gutierrez-Monreal MA, Lahens NF, Gong MC, Andrade F, Esser KA, Hughes ME | 2017 | Transcriptional Profiling Reveals Extraordinary Diversity Among Skeletal Muscle Tissues (mouse) | https://www.ncbi.nlm.nih.gov/bioproject/PRJNA391968 | NCBI BioProject, PRJNA391968 |
| Crocker A, Guan XJ, Murphy CT | 2015 | Transcriptome Analysis of *Drosophila* Mushroom Body Neurons by Cell Type Reveals Memory-Related Changes in Gene Expression | https://www.ncbi.nlm.nih.gov/bioproject/PRJNA302146 | NCBI BioProject, PRJNA302146 |
| DeAguero AA, Castillo L, Oas ST, Kiani K, Bryantsev AL, Cripps RM | 2018 | Regulation of fiber-specific actin expression by the *Drosophila* SRF ortholog Blistered | https://www.ncbi.nlm.nih.gov/bioproject/PRJNA485776 | NCBI BioProject, PRJNA485776 |

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
