## [Editor Report · eLife assessment]

This **fundamental** study reports the deep evolutionary conservation of a core genetic program regulating spermatogenesis in flies, mice, and humans. **Convincing** data were presented and supported the main conclusion. This work will be of interest to evolutionary and reproductive biologists.

---

## [Referee Report · Reviewer #1 (Public review)]

Summary:

By combining an analysis of the evolutionary age of the genes expressed in male germ cells, a study of genes associated with spermatocyte protein-protein interaction networks and functional experiments in *Drosophila*, Brattig-Correia and colleagues provide evidence for an ancient origin of the genetic program underlying metazoan spermatogenesis. This leads to the identification of a relatively small core set of functional interactions between deeply conserved gene expression regulators, whose impairment is then shown to be associated with cases of human male infertility.

Strengths:

In my opinion, the work is important for three different reasons. First, it shows that, even though reproductive genes can evolve rapidly and male germ cells display a significant level of transcriptional noise, it is still possible to obtain convincing evidence that a conserved core of functionally interacting genes lies at the basis of the male germ transcriptome. Second, it reports an experimental strategy that could also be applied to gene networks involved in different biological problems. Third, the authors make a compelling case that, due to its effects on human spermatogenesis, disruption of the male germ cell orthoBackbone can be exploited to identify new genetic causes of infertility.

Weaknesses:

The main strength of the general approach followed by the authors is, inevitably, also a weakness. This is because a study rooted in comparative biology is unlikely to identify newly emerged genes that may adopt key roles in processes such as, for example, species-specific gamete recognition. Additionally, the use of a TPM >1 threshold for protein-coding transcripts - which, as the authors pointed out, was a necessary compromise due to the high transcriptional noise of the system under study - may exclude genes, such as those encoding proteins required for gamete fusion, which are thought to be expressed at a very low level. Although these considerations raise the possibility that the chosen approach may miss information that, depending on the species, could be potentially highly functionally important, this by no means reduces its value in identifying genes belonging to the conserved genetic program of spermatogenesis. Moreover, as mentioned in the Discussion, future variations of the pipeline described in the manuscript may allow us to extend the reach of the present analysis.

---

## [Referee Report · Reviewer #2 (Public review)]

Summary:

This is a tour de force study that aims to understand the genetic basis of male germ cell development across three animal species (human, mouse and flies) by performing a genetic program conservation analysis (using phylostratigraphy and network science) with a special emphasis on genes that peak or decline during mitosis-to-meiosis. This analysis, in agreement with previous findings, reveals that several genes active during and before meiosis are deeply conserved across species, suggesting ancient regulatory mechanisms. To identify critical genes in germ cell development, the investigators integrated clinical genetics data, performing gene knockdown and knockout experiments in both mice and flies. Specifically, over 900 conserved genes were investigated in flies, with three of these genes further studied in mice. Of the 900 genes in flies, ~250 RNAi knockdowns had fertility phenotypes. The fertility phenotypes for the fly data can be viewed using the following browser link: https://pages.igc.pt/meionav. The scope of target gene validation is impressive. Below are a few minor comments.

(1) In Supplemental Figure 2, it is notable that enterocyte transcriptomes are predominantly composed of younger genes, contrasting with the genetic age profile observed in brain and muscle cells. This difference is an intriguing observation and it would be curious to hear author comments.

(2) Regarding the document, the figures provided only include supplemental data; none of the main text figures are in the full PDF.

(3) Lastly, it would be great to section and stain mouse testis to classify the different stages of arrest during meiosis for each of the mouse mutants in order to compare more precisely to flies.

This paper serves as a vital resource, emphasizing that only through the analysis of hundreds of genes can we prioritize essential genes for germ cell development. its remarkable that about 60% of conserved genes have no apparent phenotype during germ cell development.

Strengths:

High-throughput screening was conducted on a conserved network of 920 genes expressed during the mitosis-to-meiosis transition. Approximately 250 of these genes were associated with fertility phenotypes. Notably, mutations in 5 of the 250 genes have been identified in human male infertility patients. Furthermore, 3 of these genes were modeled in mice, where they were also linked to infertility. This study establishes a crucial groundwork for future investigations into germ cell development genes, aiming to delineate their essential roles and functions.

Weaknesses:

The fertility phenotyping in this study is limited, yet dissecting the mechanistic roles of these proteins falls beyond its scope. Nevertheless, this work serves as an invaluable resource for further exploration of specific genes of interest.

---

## [Author Response]

The following is the authors’ response to the original reviews.

**eLife assessment:**
This important study reports the deep evolutionary conservation of a core genetic program regulating spermatogenesis in flies, mice, and humans. The data presented are supportive of the main conclusion and generally convincing. This work will be of interest to evolutionary and reproductive biologists.

The Authors would like to thank the Senior Editor and the two Reviewers for their positive assessment of our work, as well as for the helpful suggestions. Collectively, these suggestions provided insight that was instrumental in shaping the final version of the manuscript (see below for our point-by-point comments). The Authors believe that the refinements introduced to the final document clearly translate into an improved version of our work. Hence, we would like to thank all those involved in the peer review process for their encouraging words and constructive criticism.

**Public Reviews:**

**Reviewer #1 (Public Review):**
Summary:By combining an analysis of the evolutionary age of the genes expressed in male germ cells, a study of genes associated with spermatocyte protein-protein interaction networks and functional experiments in *Drosophila*, Brattig-Correia and colleagues provide evidence for an ancient origin of the genetic program underlying metazoan spermatogenesis. This leads to identifying a relatively small core set of functional interactions between deeply conserved gene expression regulators, whose impairment is then shown to be associated with cases of human male infertility.Strengths:In my opinion, the work is important for three different reasons. First, it shows that, even though reproductive genes can evolve rapidly and male germ cells display a significant level of transcriptional noise, it is still possible to obtain convincing evidence that a conserved core of functionally interacting genes lies at the basis of the male germ transcriptome. Second, it reports an experimental strategy that could also be applied to gene networks involved in different biological problems. Third, the authors make a compelling case that, due to its effects on human spermatogenesis, disruption of the male germ cell orthoBackbone can be exploited to identify new genetic causes of infertility.

We thank the Reviewer for their positive assessment. Indeed, it was our main objective to convincingly demonstrate these three points.

Weaknesses:The main strength of the general approach followed by the authors is, inevitably, also a weakness. This is because a study rooted in comparative biology is unlikely to identify newly emerged genes that may adopt key roles in processes such as species-specific gamete recognition. Additionally, using a TPM >1 threshold for protein-coding transcripts may exclude genes, such as those encoding proteins required for gamete fusion, which are thought to be expressed at a very low level. Although these considerations raise the possibility that the chosen approach may miss information that, depending on the species, could be potentially highly functionally important, this by no means reduces its value in identifying genes belonging to the conserved genetic program of spermatogenesis.

The Authors acknowledge the points raised by the Reviewer as inevitable trade-offs of the focus of our study (to uncover the deeply conserved genetic basis of spermatogenesis). Certainly, our pipeline could, in the future, be adapted to look for newly emerged genes or to employ different minimum expression cut-offs. To this end, we made all computational data and custom scripts easily available to the community. We would, nevertheless, kindly emphasize the challenge associated with the use of less restrictive TPM cut-offs, given the substantial level of transcriptional noise associated with this cell type. An abridged version of this discussion can be found in lines 512-515 of the manuscript.

**Reviewer #2 (Public Review):**
Summary:This is a tour de force study that aims to understand the genetic basis of male germ cell development across three animal species (human, mouse, and flies) by performing a genetic program conservation analysis (using phylostratigraphy and network science) with a special emphasis on genes that peak or decline during mitosis-to-meiosis. This analysis, in agreement with previous findings, reveals that several genes active during and before meiosis are deeply conserved across species, suggesting ancient regulatory mechanisms. To identify critical genes in germ cell development, the investigators integrated clinical genetics data, performing gene knockdown and knockout experiments in both mice and flies. Specifically, over 900 conserved genes were investigated in flies, with three of these genes further studied in mice. Of the 900 genes in flies, ~250 RNAi knockdowns had fertility phenotypes. The fertility phenotypes for the fly data can be viewed using the following browser link:https://pages.igc.pt/meionav. The scope of target gene validation is impressive. Below are a few minor comments.

We thank the Reviewer for their positive appraisal of our work.

(1) In Supplemental Figure 2, it is notable that enterocyte transcriptomes are predominantly composed of younger genes, contrasting with the genetic age profile observed in brain and muscle cells. This difference is an intriguing observation and it would be curious to hear the author's comments.

Indeed, this is an intriguing observation for which we can only provide a speculative answer. Enterocytes are specialized to absorb nutrients, hence their genetic program is finely tuned to maximize uptake under specific dietary conditions. In this regard, we can posit that variations in nutrient preference/availability in the course of each species’ evolutionary history (associated with habitat, environmental and/or behavioral changes) may have exerted a selective pressure for the emergence of new genes that could provide enterocytes with more efficient uptake capabilities under new circumstances. The application of evolutionary thinking to the rapidly expanding field of nutrigenomics could shed light on this possibility.

(2) Regarding the document, the figures provided only include supplemental data; none of the main text figures are in the full PDF.

We thank the Reviewer for this helpful comment. We will ensure that the three main figures are correctly formatted in the final version of the manuscript.

(3) Lastly, it would be great to section and stain mouse testis to classify the different stages of arrest during meiosis for each of the mouse mutants in order to compare more precisely to flies.

We agree with the Reviewer that adding more mouse data would further improve what can already be considered an extensive body of experimental work. Given the costs associated with the generation of such data (in terms of resources and otherwise), the Authors believe such a study would be best suited to a follow-up manuscript.

This paper serves as a vital resource, emphasizing that only through the analysis of hundreds of genes can we prioritize essential genes for germ cell development. its remarkable that about 60% of conserved genes have no apparent phenotype during germ cell development.

Once again, we thank the Reviewer for their positive assessment of our work. Clarifying the degree of functional redundancy in an essential biological process such as male gametogenesis represents an exciting (and experimentally complex) future challenge.

Strengths:The high-throughput screening was conducted on a conserved network of 920 genes expressed during the mitosis-to-meiosis transition. Approximately 250 of these genes were associated with fertility phenotypes. Notably, mutations in 5 of the 250 genes have been identified in human male infertility patients. Furthermore, 3 of these genes were modeled in mice, where they were also linked to infertility.This study establishes a crucial groundwork for future investigations into germ cell development genes, aiming to delineate their essential roles and functions.

The Authors thank the Reviewer for emphasizing the potential usefulness of our results to the community, as that was one of the main motivations behind this project.

Weaknesses:The fertility phenotyping in this study is limited, yet dissecting the mechanistic roles of these proteins falls beyond its scope. Nevertheless, this work serves as an invaluable resource for further exploration of specific genes of interest.

Please see the previous point.

**Recommendations for the authors:**

**Reviewer #1 (Recommendations For The Authors):**
Although the manuscript already includes a significant amount of data, there are two aspects that the authors may consider exploring:(1) I understand that the choice of species whose gene expression was analyzed in the study was largely influenced by the quality of the corresponding genome annotations. However, since in evolutionary terms humans and mice are much closer to each other than *Drosophila* (as also shown in Figure 1c and Supplementary Figure 1), I found the statement "three evolutionarily distant gonochoric species" partially questionable. Have the authors considered adding an additional established animal model, such as for example zebrafish, to provide further coverage of the evolutionary space? Or, alternatively, could a posteriori analysis of the transcriptome of such an additional species be used to cross-validate their findings? The authors touch upon this point in the Discussion, but I wonder if they actually tried something in this direction, or simply decided that the currently available expression data from other organisms was too poor to be used for this purpose.

We thank the Reviewer for bringing up this point, as it echoes one of our main concerns in terms of our approach (as discussed in lines 487-492). Indeed, when we were designing our study, we extensively discussed whether zebrafish and *C. elegans* datasets should be included, as high-quality expression and phenotypical data were available for both species. We ended up not including them for one main reason: the sexual system of these species deviates from that of humans, mice and fruit flies (all gonochoric species). More specifically, *C. elegans* are hermaphrodites and although zebrafish is a gonochoric species at the adult stage, they start their lifecycle as juvenile hermaphrodites (they first develop juvenile ovaries that later degenerate into a testis in males). Since it is largely unknown to what extent the transcriptome of male germ cells from these species deviates from the gonochoric program (by retaining oogenesis-related characteristics, for example), we decided to avoid possible confounding effects by excluding the two species. Undoubtedly, as more transcriptomic data from non-model organisms become available, these (and other) questions can be extensively revisited as our pipeline was designed to easily accommodate new data.

(2) Although the use of the STRING database is a sensible choice given the general purpose of this work, in my experience the reliability of its individual interactions can vary significantly. I wonder if the authors have considered exploiting AlphaFold-Multimer as a parallel approach to estimate what proportion of the 79 functional interactions that they identified may reflect direct protein-protein contacts.

We thank the Reviewer for this question and suggestion, as we were also concerned about STRING's reliability for individual interactions. For thatreason, we only utilized protein-protein interactions with a STRING combined confidence score ≥0.5(corresponding to the estimated likelihood of a given association beingtrue), as described in more detail in the "Protein-protein interaction(PPI) network construction" subsection. In addition, to make sure we were not biasing results towards conserved genes (which could arguably be overrepresented in STRING) we pursued a random rewiring test of degreecentrality and page rank, as detailed in section "Deeply conserved genesare central components of the male germ cell transcriptome". We very much like the suggestion of using AlphaFold-Multimer to estimate the proportion ofdirect protein-protein contacts for the 79 core interactions, but giventhe already quite complex analytical pipeline of the present work, we will leave such analysis for a follow-up study. The final version of the manuscript now contains a reference to such an approach (lines 499-502).

Finally, probably because my primary focus is not on gene regulation, I must say that I found the manuscript somewhat heavy to read. The integration of various data types and analyses, while enriching, also complicates the ability to clearly recall the main conclusions of each result section by the time one reaches the summary at the beginning of the Discussion. Given the relative brevity of the latter, expanding it to both reiterate what these conclusions are and illustrate how all the components converge to support the central message of the study would, in my opinion, benefit a general readership.

We thank the Reviewer for their fresh perspective on our document and for this most welcome suggestion. The final version of the manuscript now includes a longer discussion, containing an initial paragraph (lines 467-479) that summarizes our main findings and how they converge into a coherent body of work.

Additionally, on a minor note, I suggest that the concept of phylostratigraphy be briefly explained when first mentioned in the Introduction, rather than later in the manuscript. This early clarification would aid comprehension for readers unfamiliar with the term.

To safeguard the flow of the manuscript, we have slightly tweaked the introduction section to avoid the use of highly specific terminology (such as phylostratigraphy) this early in the text. We replaced it with “comparison of genome sequences” (line 85). Phylostratigraphy is later explained in full detail in the corresponding section of the manuscript. We thank the Reviewer for this helpful suggestion.

**Reviewer #2 (Recommendations For The Authors):**
Major concern - the absence of main text figures.

We thank the Reviewer for this helpful comment. We will ensure that the three main figures are correctly formatted in the final version of the manuscript.

Typos throughout - this will need your attention.

The Authors thank the Reviewer for the thorough and attentive assessment of our work. We have carefully revised the text to ensure a pleasant reading experience free of typographical errors.